# Relative Translation Invariant Wasserstein Distance

**Binshuai Wang**                                          *derekwang@gwu.edu*
*Department of Computer Science*
*George Washington University*

**Qiwei Di**                                               *qiwei2000@cs.ucla.edu*
*Department of Computer Science*
*University of California, Los Angeles*

**Ming Yin**                                               *my0049@princeton.edu*
*Department of Electrical and Computer Engineering*
*Princeton University*

**Mengdi Wang**                                            *mengdiw@princeton.edu*
*Department of Electrical and Computer Engineering*
*Princeton University*

**Quanquan Gu**                                            *qgu@cs.ucla.edu*
*Department of Computer Science*
*University of California, Los Angeles*

**Peng Wei**                                               *pwei@gwu.edu*
*Department of Computer Science*
*George Washington University*

**Reviewed on OpenReview:** *https: // openreview. net/ forum? id= NfhVTi2G4a*

## Abstract

Motivated by the Bures distance, we introduce a new family of distances, *relative translation invariant Wasserstein distances*, denoted by $RW_p$, as an extension of the classical Wasserstein distances $W_p$ for $p \in [1, +\infty)$. We establish that $RW_p$ defines a valid metric and demonstrate that this type of metric is more intrinsic than the classical Wasserstein distance. A bi-level algorithm is designed to compute the general $RW_p$ distance between arbitrary discrete distributions. Moreover, when $p = 2$, we show that the optimal coupling matrix is invariant under distributional translation in the discrete setting, and we further propose two algorithms, the $RW_2$-LP algorithm and the $RW_2$-Sinkhorn algorithm, to improve the numerical stability of computing $W_2$ distance and the optimal coupling matrix solutions. Finally, we conduct three experiments to validate our theoretical results and algorithms. The first two experiments report that the $RW_2$-LP algorithm and the $RW_2$-Sinkhorn algorithm, both with and without normalization, can significantly reduce the numerical errors compared to standard algorithms. The third experiment shows that $RW_p$ algorithms are computationally scalable and applicable to the retrieval of similar thunderstorm patterns in practical applications. The implementation is publicly available at `https://github.com/DRKWang/rw_metric`.

## 1 Introduction

Optimal transport (OT) theory provides a rigorous and interpretable framework for measuring discrepancies between probability distributions. Due to its strong theoretical foundations and flexibility, OT has become one of the central tools in modern machine learning. It has found wide-ranging applications in domain adaptation (Courty et al., 2017), generative modeling—most notably in Wasserstein GANs (Arjovsky et al., 2017)—and

evaluation metrics such as the Fréchet Inception Distance (FID) (Heusel et al., 2017). In addition, OT also played an important role in distributionally robust learning, including regression (Shafieezadeh-Abadeh et al., 2015; Chen & Paschalidis, 2018) and Markov decision processes (Yu et al., 2023), as well as in object tracking and matching using graph neural networks (Sarlin et al., 2019; Grand-Clément & Kroer, 2021).

Although many computational methods, such as linear programming–based solvers (Villani, 2009; Peyré & Cuturi, 2019) and the Sinkhorn algorithm (Cuturi, 2013), which have been developed to compute optimal transport efficiently, practical data settings can still lead to a loss of precision. Measurement errors and systematic perturbations are inevitable in real-world settings. When two distributions are very close, it becomes difficult to distinguish whether an observed discrepancy reflects intrinsic differences in the underlying data or arises from exogenous factors such as sensor noise, calibration drift, or other systematic biases. While modern OT methods can accurately quantify distributional differences, their sensitivity to such perturbations may lead to instability in downstream tasks and hinder robust performance in practice. As a result, it is natural to raise the following question:

*Can we design a new metric, along with an efficient algorithm, that captures intrinsic differences between probability distributions regardless of systematic perturbations?*

To answer this question, we introduce a new family of distances, *relative translation invariant Wasserstein distances*[1] ($RW_p$), as an extension of the classical Wasserstein distances $W_p$ for $p \in [1, +\infty)$. Compared with the classical Wasserstein distances, these new distances are more robust to systematic perturbations and global translational shifts. We propose an efficient algorithm to compute the general $RW_p$ distances. In the special case $p = 2$, we show that the optimal transport coupling matrix solution is invariant under any relative translation. Building on this property, we further develop two algorithms, the $RW_2$-LP algorithm and the $RW_2$-Sinkhorn algorithm, to reduce computational errors and improve numerical stability. Finally, we conduct three experiments to validate the effectiveness of the proposed algorithms. The first two experiments demonstrate that the $RW_2$-LP algorithm and $RW_2$-Sinkhorn algorithm can improve numerical stability compared to standard algorithms. The third experiment validates that the $RW_p$ algorithms are computationally scalable and applicable to similar thunderstorm retrieval in real-world applications.

**Contributions.** The main contributions of this paper are summarized as follows:

*(a)* We introduce a new family of distances, *relative translation invariant Wasserstein distances* ($RW_p$), and prove that they are true metrics and invariant to relative translations of probability distributions.

*(b)* We design a bi-level algorithm for efficiently computing the general $RW_p$ distances between arbitrary discrete distributions for arbitrary $p \geq 1$.

*(c)* We show that, in discrete settings when $p = 2$, the optimal coupling matrix is invariant under distributional translations. Based on this property, we develop two algorithms for the LP-based optimal transport algorithm and the Sinkhorn algorithm to improve the numerical stability in the computation of the $W_2$ distance. In particular, we show that the $RW_2$-Sinkhorn algorithm offers improved numerical stability while maintaining the same convergence rate as the standard Sinkhorn algorithm. Our experiments also report that the $RW_2$-LP algorithm and the $RW_2$-Sinkhorn algorithm, both with and without normalization, can significantly reduce numerical errors.

*(d)* We demonstrate the practical applications of $RW_p$ distances in the tasks of retrieval of similar thunderstorm patterns, showcasing their effectiveness in large-scale real-world applications.

**Organization.** The remainder of the paper is organized as follows. Section 2 reviews classical results in optimal transport theory, Bures distance and the Sinkhorn algorithm. Section 3 provides the definition of the $RW_p$ distances and some key properties of the distances. Section 4 presents computational algorithms for the general $RW_p$ distances and the $RW_2$-based algorithms for the LP-based OT algorithm and the Sinkhorn algorithm, along with an analysis of their stability and convergence rate. Finally, Section 5 provides numerical

---

[1]We use the term *relative* to distinguish this notion from the basic translation-invariance property of Wasserstein distances, namely $W_p(\mu + t, \nu + t) = W_p(\mu, \nu), \forall t \in \mathbb{R}^d$. More details of this property can be found in Corollary 1.16 of Villani (2003).

validation for the $RW_2$-LP algorithm and the $RW_2$-Sinkhorn algorithm and demonstrates the retrieval results of similar thunderstorm patterns.

**Notations.** Let $\mathcal{P}_p(\mathbb{R}^d)$ denote the set of all probability distributions on $\mathbb{R}^d$ with finite $p$th-order moments. For simplicity, we let $\mu$ and $\nu$ denote a pair of source and target distributions, respectively. $\mu$ and $\nu$ are supported on finite sets $\{x_i\}_{i=1}^{n_1}$ and $\{y_j\}_{j=1}^{n_2}$, respectively, where $n_1$ and $n_2$ denote the numbers of support points. $\bar{\mu}$ and $\bar{\nu}$ are the means (mass centers) of $\mu$ and $\nu$, respectively. Let $\mathbb{R}_*^{n_1 \times n_2}$ be the set of all $n_1 \times n_2$ matrices with non-negative entries. We use $[\mu]$ to denote the equivalence class (orbit) of $\mu$ under the translation equivalence relation in $\mathcal{P}_p(\mathbb{R}^d)$. $\mathbb{S}_+^d$ denotes the set of $d \times d$ symmetric positive semidefinite matrices. The vector $\mathbf{1}$ denotes the all-ones vector. The operation ./ denotes the component-wise vector division. $\|\cdot\|$ denotes a norm on $\mathbb{R}^d$. $\|C\|_\infty$ denotes the value of the largest component in matrix $C$.

**Related work.** Several works have investigated variants of OT that incorporate transformation invariance in the underlying space (Khesin et al., 2011; Alvarez-Melis et al., 2019; Adamo et al., 2025). For example, Alvarez-Melis et al. (2019) study optimal transport with global invariance, where the transport cost is minimized jointly over transport plans and transformations of the distributions. Their formulation provides a general framework for invariant OT problems and includes translation-invariant transport as a special case. Related directions include invariant and structure-aware OT formulations that compare objects modulo transformations or relational structures. Our work focuses specifically on translation invariance and studies the resulting quotient metric structure together with its computational implications.

Another related line of work is the Gromov–Wasserstein distance ($GW$) or Entropy-regularized Gromov–Wasserstein distance ($EGW$) (Mémoli, 2011; Séjourné et al., 2021; Peyré et al., 2016; Xu et al., 2019), which compares the relational structures of two metric measure spaces. When Euclidean distances are used, the $GW$ formulation can exhibit invariance to global translations and rotations. However, $GW$ focuses on matching pairwise structural relations between points, whereas our approach directly aligns distributions under relative translations while preserving the classical OT coupling structure. Moreover, $GW$ does not define a true metric, while our method is formulated as a proper metric. In this work, we also include both $GW$ and $EGW$ as baselines for similar thunderstorm pattern retrievals in Subsection 5.2.

There is also a growing body of work on robust optimal transport formulations. These methods aim to mitigate the effect of outliers, noise, or adversarial perturbations in the input distributions by modifying the transport objective (Balaji et al., 2020; Mukherjee et al., 2021).

The Bures distance (Bhatia et al., 2019; van Oostrum, 2020) is another closely related concept. It arises as the covariance component of the 2-Wasserstein distance between Gaussian measures and provides a Riemannian metric on the cone of positive semidefinite matrices. However, this distance is restricted to Gaussian distributions. In contrast, our formulation extends this perspective beyond the Gaussian setting by considering general probability distributions with finite moments and general $p$-norm transport costs.

Finally, Appendix C introduces a rotation-invariant extension obtained by minimizing the transport cost over rotations, also known as the Procrustes–Wasserstein distance (Zhang et al., 2017; Grave et al., 2018; Adamo et al., 2025). This construction is also related to invariant OT formulations and connects to ideas explored in sliced Gromov–Wasserstein methods (Vayer et al., 2022) and sliced optimal transport on spheres (Quellmalz et al., 2024), where invariances are incorporated through transformations of the underlying geometry.

## 2 Preliminaries

Before presenting our proposed method, we briefly review key concepts and formulations from classical optimal transport theory. This section establishes the technical foundation for Section 3.

### 2.1 Optimal Transport Theory

Optimal transport (OT) addresses the problem of finding a minimal-cost plan for transporting one probability distribution to another on a metric space. Given a cost function $c(x, y)$ and two probability measures $\mu(x)$ and $\nu(y)$ on $\mathbb{R}^d$, the goal is to identify a transport plan between $\mu$ and $\nu$ that minimizes the total cost

of moving masses from $\mu$ to $\nu$ under $c(x, y)$. While the cost function can be any non-negative function, a common and particularly useful choice is a distance-based cost of order $p$, such as $c(x, y) = \|x - y\|^p$, where $\|\cdot\|$ denotes a norm on $\mathbb{R}^d$ and $p \in [1, \infty)$. Under these mild conditions, the corresponding OT problem is well-defined (Villani, 2003).

Let $\mu(x)$ be the source distribution and $\nu(y)$ the target distribution, with $\mu, \nu \in \mathcal{P}_p(\mathbb{R}^d)$. The optimal transport problem can be formulated as the following optimization problem.

**Definition 2.1** ($p$-norm optimal transport problem (Villani, 2003))**.**

$$\mathrm{OT}(\mu, \nu, p) := \min_{\gamma \in \Gamma(\mu, \nu)} \int_{\mathbb{R}^{2d}} \|x - y\|^p \, d\gamma(x, y), \tag{1}$$

where

$$\Gamma(\mu, \nu) = \left\{ \gamma \in \mathcal{P}_p(\mathbb{R}^{2d}) \, \Big| \, \int_{\mathbb{R}^d} \gamma(x, y) \, dy = \mu(x), \, \int_{\mathbb{R}^d} \gamma(x, y) \, dx = \nu(y) \right\}.$$

Here, $\gamma(x, y)$ is a transport plan (or coupling), specifying how much mass is moved from source location $x$ to target location $y$. The objective is to minimize the total transport cost, i.e., the overall cost of moving masses across all source–target pairs $(x, y)$.

Building on this formulation, one obtains a family of distances on $\mathcal{P}_p(\mathbb{R}^d)$ known as Wasserstein distances (Villani, 2009), defined by the optimal transport cost. It is worth noting that the norm $\|\cdot\|$ can have a different order from the order $p$.

**Definition 2.2** ($p$-Wasserstein distance (Villani, 2009))**.** The $p$-Wasserstein distance between two probability distributions $\mu$ and $\nu$ is given by

$$W_p(\mu, \nu) := \mathrm{OT}(\mu, \nu, p)^{1/p}, \quad p \in [1, \infty).$$

The Wasserstein distance defines a true metric on $\mathcal{P}_p(\mathbb{R}^d)$, satisfying non-negativity, identity of indiscernibles, symmetry, and the triangle inequality (Villani, 2009). Moreover, it is well-defined for a broad type of probability distributions, including both discrete and continuous distributions.

In practical applications, the functional optimization in Equation 1 is typically reformulated as a discrete optimization problem. In this setting, the distributions $\mu$ and $\nu$ are represented by a *finite* number of support points (data samples) $\{x_i\}_{i=1}^{n_1}$ and $\{y_j\}_{j=1}^{n_2}$, with associated probability masses $\{a_i\}_{i=1}^{n_1}$ and $\{b_j\}_{j=1}^{n_2}$, where $n_1$ and $n_2$ denote the number of support points, respectively.

Since both $n_1$ and $n_2$ are finite, we define a cost matrix $C \in \mathbb{R}_*^{n_1 \times n_2}$, whose entries represent the transport cost from $x_i$ to $y_j$,

$$C_{ij} = \|x_i - y_j\|^p.$$

The discrete optimal transport problem can then be regarded as a linear program

$$\mathrm{OT}(\mu, \nu, p) = \min_{P \in \Pi(\mu, \nu)} \sum_{i=1}^{n_1} \sum_{j=1}^{n_2} C_{ij} \, P_{ij}, \tag{2}$$

where the feasible set is

$$\Pi(\mu, \nu) = \left\{ P \in \mathbb{R}_*^{n_1 \times n_2} \, \Big| \, P\mathbf{1} = \mathbf{a}, \, P^\top \mathbf{1} = \mathbf{b} \right\}.$$

Here, $P_{ij}$ denotes the coupling variable, representing the amount of transported mass from source point $x_i$ to target point $y_j$. This linear programming formulation provides a tractable and widely used approach for solving discrete OT problems in practical applications.

## 2.2 Bures Distance for Gaussian Distributions

The 2-Wasserstein distance ($W_2$) admits a closed-form expression when both probability distributions are Gaussian distributions. This result leads to the *Bures distance*[2], which plays an important role in the geometry of Gaussian distributions.

Let $\mu = \mathcal{N}(\bar{\mu}, \Sigma_\mu)$ and $\nu = \mathcal{N}(\bar{\nu}, \Sigma_\nu)$ be two Gaussian distributions in $\mathbb{R}^d$ with means $\bar{\mu}, \bar{\nu} \in \mathbb{R}^d$ and covariance matrices $\Sigma_\mu, \Sigma_\nu \in \mathbb{S}_+^d$, where $\mathbb{S}_+^d$ denotes the set of $d \times d$ symmetric positive semidefinite matrices. The squared 2-Wasserstein distance between $\mu$ and $\nu$ has the closed-form expression (Olkin & Pukelsheim, 1982; Dowson & Landau, 1982; Givens & Shortt, 1984; Knott & Smith, 1984)

$$W_2^2(\mu, \nu) = \|\bar{\mu} - \bar{\nu}\|_2^2 + \mathrm{Tr}(\Sigma_\mu) + \mathrm{Tr}(\Sigma_\nu) - 2\,\mathrm{Tr}\left((\Sigma_\mu^{1/2}\Sigma_\nu\Sigma_\mu^{1/2})^{1/2}\right).$$

This expression naturally decomposes the Wasserstein distance into a contribution from the means and a contribution from the covariance matrices. The covariance term

$$d_B(\Sigma_\mu, \Sigma_\nu) = \sqrt{\mathrm{Tr}(\Sigma_\mu) + \mathrm{Tr}(\Sigma_\nu) - 2\,\mathrm{Tr}\left((\Sigma_\mu^{1/2}\Sigma_\nu\Sigma_\mu^{1/2})^{1/2}\right)}$$

is known as the *Bures distance* between covariance matrices (Bhatia et al., 2019; Peyré & Cuturi, 2019; van Oostrum, 2020).

This distance is closely related to the Bures distance studied in quantum information geometry and defines a Riemannian metric on the manifold of symmetric positive semidefinite matrices (Bures, 1969; Bhatia et al., 2019; Peyré & Cuturi, 2019). In particular, for Gaussian distributions, the squared 2-Wasserstein distance decomposes as

$$W_2^2(\mu, \nu) = \|\bar{\mu} - \bar{\nu}\|_2^2 + d_B^2(\Sigma_\mu, \Sigma_\nu). \tag{3}$$

This equation highlights that the Bures distance captures the covariance component of Gaussian distributions under the Wasserstein geometry. Moreover, the $RW_2$ distance introduced in this work extends this perspective beyond the Gaussian setting. Specifically, while the Bures distance characterizes the distance between covariance matrices of centered Gaussian measures, the $RW_2$ distance generalizes the centered component of the $W_2$ distance to arbitrary probability distributions with finite second moments. More detailed discussions can be found in Subsection 3.3.

## 2.3 Sinkhorn Algorithm

The discrete OT problem in Equation 2 is a linear program that can be solved by simplex or interior-point algorithms (Peyré & Cuturi, 2019). However, for large-scale problems, these approaches can become computationally expensive. A popular alternative approach exploits the special structure of the feasible set $\Pi(\mu, \nu)$ by introducing a (negative) entropy regularization term in the objective function (Cuturi, 2013). This leads to a strictly convex optimization problem whose solution can be obtained via a simple matrix scaling procedure known as the Sinkhorn algorithm.

The (negative) entropy-regularized OT problem is given by

$$\mathrm{OT}_\lambda(\mu, \nu, p) := \min_{P \in \Pi(\mu,\nu)} \sum_{i,j} C_{ij}P_{ij} + \lambda \sum_{i,j} P_{ij}(\log P_{ij} - 1),$$

where $\lambda > 0$ controls the strength of the regularization. Defining

$$K_{ij} = \exp\left(-\frac{C_{ij}}{\lambda}\right),$$

---

[2]The Bures distance is the Riemannian metric on the manifold of symmetric positive-definite covariance matrices and coincides with the 2-Wasserstein distance between mean-zero Gaussian distributions. The term *Bures—Wasserstein* distance is sometimes used to emphasize this connection with optimal transport (van Oostrum, 2020). However, when non-centered Gaussian distributions are considered, this terminology may lead to ambiguity, since the full Wasserstein distance additionally includes the Euclidean distance between the means. In this work, we therefore use the term Bures distance to refer specifically to the covariance component, and Wasserstein distance between Gaussian distributions for the full transport distance, in order to avoid ambiguity.

the optimal coupling can be written in the factorized form $P = \operatorname{diag}(u)\, K \operatorname{diag}(v)$ for some positive scaling vectors $u \in \mathbb{R}^{n_1}$ and $v \in \mathbb{R}^{n_2}$ satisfying the marginal constraints.

The Sinkhorn algorithm starts from initial vectors $u^{(0)} = v^{(0)} = \mathbf{1}$. For iteration $k \geq 0$, the updates proceed alternately as
$$u^{(k+1)} \leftarrow a./(Kv^{(k)}), \quad v^{(k+1)} \leftarrow b./(K^\top u^{(k+1)}),$$
where the division is component-wise.

Once the updates converge to the optimal $(u^*, v^*)$, the coupling matrix $P$ can be recovered as
$$P = \operatorname{diag}(u^*)\, K \operatorname{diag}(v^*).$$

As $\lambda \to 0$, the entropy-regularized optimal transport solution converges to the optimal solution of the original OT linear program (Cuturi, 2013; Peyré & Cuturi, 2019), while for fixed $\lambda > 0$ the Sinkhorn iterations are computationally efficient and scalable.

## 3 Relative Translation Optimal Transport and $RW_p$ Distances

In this section, we introduce the *relative translation optimal transport* (ROT) problem and provide the definition of *relative translation invariant Wasserstein distance* ($RW_p$). We establish basic properties, including the existence of the minimizers and $RW_p$ defines a true metric for $p \geq 1$. Special attention is devoted to the quadratic case ($p = 2$), where additional structure allows that the optimal coupling matrix is invariant under translations.

### 3.1 Relative Translation Optimal Transport and the $RW_p$ Distance

Classical optimal transport problem compares two distributions in a fixed coordinate system. However, when the primary difference between two distributions is caused by a global translation of their support points, the classical OT distance may overestimate the global translation, rather than their intrinsic difference. To measure the intrinsic difference, we introduce the *relative translation optimal transport* problem, which aligns one distribution with the other through a *relative* coordinate system, rather than a fixed coordinate system.

**Definition 3.1** (Relative translation optimal transport problem). Let $\mu, \nu \in \mathcal{P}_p(\mathbb{R}^d)$. The relative translation optimal transport problem is defined as
$$\operatorname{ROT}(\mu, \nu, p) := \inf_{t \in \mathbb{R}^d} \operatorname{OT}(\mu + t, \nu, p), \tag{4}$$
where $t \in \mathbb{R}^d$ is a translation vector and $(\mu + t)$ denotes the pushforward of $\mu$ under the map $x \mapsto x + t$.

This formulation introduces an outer optimization over $t$, while the inner optimization corresponds to the classical $p$-Wasserstein problem in terms of the translated distribution $\mu + t$. As a result, the ROT problem captures the minimal transport cost while dynamically aligning the two distributions.

The following proposition shows that the search domain for the optimal translation can be restricted to a compact set. Therefore, the minimizer exists, and the minimal value can be achieved.

**Proposition 3.2** (Compactness and existence of minimizers). *In 4, the search for the optimal translation $t$ may be restricted to the following compact ball set*
$$B = \left\{ t \in \mathbb{R}^d : \|t\| \leq 2\, W_p(\mu, \nu) \right\}.$$
*Consequently,*
$$\operatorname{ROT}(\mu, \nu, p) = \min_{t \in B} \operatorname{OT}(\mu + t, \nu, p),$$
*and the minimizer can be attained.*

The proof is provided in Appendix A.1. The compactness ensures that the minimizer of the ROT problem exists and avoids pathological behavior such as unbounded translations.

**A quotient-space perspective.** Let $\sim_T$ denote the equivalence relation on $\mathcal{P}_p(\mathbb{R}^d)$ induced by translations: we write $\mu \sim_T \mu'$ when $\mu'$ is obtained by applying a translation from $\mu$. This relation partitions $\mathcal{P}_p(\mathbb{R}^d)$ into different equivalence classes $[\mu]$, which are the elements in the quotient space

$$\mathcal{P}_p(\mathbb{R}^d)/\sim_T.$$

From this perspective, the ROT problem naturally becomes an optimal transport problem on the quotient space, whose objective is to compute the minimal transport cost between two equivalence classes $[\mu]$ and $[\nu]$.

Coming from this observation, we introduce a new family of Wasserstein distances that quantify the minimal transport cost in terms of the above translation equivalence classes of probability distributions. Since the value of the ROT problem depends only on the equivalence classes themselves, and the value is actually *invariant* under relative translations, we refer to these distances as *relative translation invariant Wasserstein distances*, denoted by $RW_p$.

**Definition 3.3** (*p*-relative translation invariant Wasserstein distance)**.** For $p \in [1, \infty)$, the relative translation invariant Wasserstein distance between equivalence classes $[\mu]$ and $[\nu]$ is

$$RW_p([\mu], [\nu]) := \mathrm{ROT}(\mu, \nu, p)^{1/p},$$

where any representatives $\mu$ and $\nu$ from $[\mu]$ and $[\nu]$ may be chosen.

The following theorem establishes that $RW_p$ is a true metric.

**Theorem 3.4.** *For any $p \in [1, \infty)$, the function $RW_p$ defines a real metric on the quotient space $\mathcal{P}_p(\mathbb{R}^d)/\sim_T$.*

The proof is provided in Appendix A.2. We remark that analogous definitions can also be made for other transformations, such as rotation (see Appendix C.1 for details). However, it is worth noting that the corresponding optimization problem for rotation is generally non-convex and difficult to solve for the global minimizers, as illustrated in the example in Appendix C.2. Because of this, we primarily focus on translation transformation in this work.

**Choice of $p$ on noise tolerance.** The choice of $p$ in the $RW_p$ distance directly influences the sensitivity of the distance to noise, in a manner similar to the $\ell_p$ distance or $W_p$ distance. Distances with smaller $p$ (e.g., $RW_1$) tend to be more robust to outliers and localized noise, since the cost grows slowly with displacement and therefore does not heavily penalize large but sparse deviations. In contrast, distances with larger $p$ (e.g., $RW_2$ or $RW_4$) amplify the influence of large transport displacements, making the distance more sensitive to outliers but simultaneously more responsive to global geometric differences in shape. Thus, different choices of $p$ imply different notions of similarity: small $p$ favors robustness, while large $p$ emphasizes shape similarity. The experimental results in Subsection 5.2 are also consistent with the above analysis.

### 3.2 Computational Tractability of the ROT Problem

The computational tractability of the ROT problem depends on the dimension of the underlying space and the structure of the distributions. In the one-dimensional case ($d = 1$), the ROT problem admits an analytical characterization due to the monotone transport structure of optimal transport plans implied by cyclical monotonicity. Specifically, using the quantile formulation of the $p$-Wasserstein distance, the optimal transport plan is given by the monotone rearrangement between the distributions (Villani, 2009; Peyré & Cuturi, 2019). This reduces the ROT problem to a one-dimensional convex optimization problem with respect to the translation variable, allowing the optimal translation to be characterized explicitly. For completeness, we provide a detailed derivation and explicit formulation in Appendix A.4. For the rest of the paper, we mainly focus on the case $d \geq 2$, where the problem becomes substantially more challenging.

When the dimension satisfies $d \geq 2$, although the ROT formulation is well-defined and admits at least one minimizer, we find the corresponding optimization problem is generally non-convex. Several illustrative examples of this non-convex behavior are provided in Appendix B. The non-convexity arises from the bilinear structure of the objective function for both the translation variable $t$ and the transport plan $P$ in Equation 4.

Nevertheless, certain special cases admit closed-form solutions. In particular, when both distributions are Gaussian, ROT problem can be solved explicitly via the Bures distance, given by

$$RW_2^2(\mu, \nu) = \operatorname{Tr}(\Sigma_\mu) + \operatorname{Tr}(\Sigma_\nu) - 2\operatorname{Tr}\left((\Sigma_\mu^{1/2}\Sigma_\nu\Sigma_\mu^{1/2})^{1/2}\right).$$

Further details are provided in Subsections 2.2 and 3.3.

For general distributions in dimensions $d \geq 2$, we solve the ROT problem using an alternating minimization scheme. Although the overall problem is non-convex, each subproblem has a tractable structure:

- For fixed $P$, the optimization with respect to the translation variable $t$ is convex.

- For fixed $t$, the optimization with respect to $P$ reduces to a classical optimal transport problem, which can be solved as a linear program.

This structure naturally motivates an alternating optimization procedure that iteratively updates $t$ and $P$. In practice, this approach converges to a local minimizer and produces stable solutions, while each update step remains computationally simple. In our implementation, we combine this scheme with dual-simplex–based reinitialization and Armijo backtracking strategies to further improve computational efficiency. More details and convergence discussions are provided in Subsection 4.1.

### 3.3 Quadratic ROT and Properties of the $RW_2$ Distance

As discussed previously, the ROT problem is non-convex in general and does not admit a decomposable structure. However, when the cost is the squared Euclidean distance, corresponding to the quadratic case $p = 2$, the problem exhibits a special structure. In particular, the squared 2-Wasserstein distance admits a classical decomposition into a term involving the means and a term corresponding to the centered distributions (see Remark 2.19 in Peyré & Cuturi, 2019). This observation leads to the following result of the quadratic ROT problem.

**Proposition 3.5** (Decomposition of the quadratic ROT)**.** *For any $\mu, \nu \in \mathcal{P}_2(\mathbb{R}^d)$, the quadratic ROT satisfies*

$$\operatorname{ROT}(\mu, \nu, 2) = \min_{t \in \mathbb{R}^d} \operatorname{OT}(\mu + t, \nu, 2) = \operatorname{OT}(\mu, \nu, 2) - \|\bar{\mu} - \bar{\nu}\|_2^2,$$

*where $\bar{\mu}$ and $\bar{\nu}$ denote the means of $\mu$ and $\nu$. Moreover, in the discrete setting, the optimal coupling matrix $P \in \mathbb{R}_*^{n_1 \times n_2}$ is invariant under any relative translation of the distributions.*

The first sentence of this proposition has been clearly discussed in Remark 2.19 of Peyré & Cuturi (2019). Essentially, this result is consistent with the classical identity for the squared 2-Wasserstein distance, which decomposes into the squared Euclidean distance between the means and the squared 2-Wasserstein distance between the centered distributions. From the ROT perspective, the optimal translation in the quadratic case is precisely the difference of the means $\bar{\mu} - \bar{\nu}$, which corresponds to centering both distributions. For completeness, we also provide a full proof of Proposition 3.5 in Appendix A.3.

This decomposition has two important implications in the discrete setting. First, in the quadratic case, the classical OT formulation and the ROT problem share the same optimal coupling matrix $P$. Second, the optimal coupling $P$ is *invariant* under any relative translation of the distributions. As a result, any representatives $\mu' \in [\mu]$ and $\nu' \in [\nu]$ from their respective translation equivalence classes yield the same optimal coupling matrix. This observation provides the theoretical foundation for the $RW_2$ algorithms based on both LP-based optimal transport solvers and the Sinkhorn algorithm introduced in Subsection 5.1.2. In practice, this invariance allows us to choose numerically convenient representatives, which can improve numerical stability. The experimental results in Subsection 5.1 further demonstrate that this decomposition can significantly reduce numerical errors.

**Centering vs. Normalization (scaling).** It is worth noting that the relative translation (or centering) operation considered in this work differs from the normalization operation commonly used in practice, which

rescales the coefficient matrix by dividing its maximum component and subsequently restores the original scale using the same factor Tomlin (1975); Peyré & Cuturi (2019). In particular, centering is an additive transformation, whereas normalization is multiplicative. Since both operations may improve numerical stability, we also include normalization as a baseline in our experiments to enable a comprehensive evaluation. Our results indicate that, under certain settings, centering can be more effective than normalization in reducing numerical errors. Further discussion is provided in Subsection 5.1.

**Corollary 3.6** (Decomposition of $W_2$ distance). *For any $\mu, \nu \in \mathcal{P}_2(\mathbb{R}^d)$,*

$$W_2^2(\mu, \nu) = \|\bar{\mu} - \bar{\nu}\|_2^2 + RW_2^2([\mu], [\nu]).$$

Equation 3 and Corollary 3.6 suggest that the $RW_2$ distance is consistent with the classical Bures distance for Gaussian distributions (Bhatia et al., 2019; Peyré & Cuturi, 2019); see Subsection 2.2 for the definition of the Bures distance. In particular, for Gaussian distributions, the Wasserstein distance between the centered components reduces to the Bures distance between their covariance matrices. The $RW_2$ distance generalizes this perspective beyond the Gaussian setting by capturing the centered component of Wasserstein geometry for general probability distributions in space $\mathcal{P}_2(\mathbb{R}^d)$.

Finally, although the above decomposition shows that the optimal translation coincides with the difference of the means when $p = 2$, this property does not necessarily hold for other orders $p$. A counterexample for this is provided in Appendix B.3.

## 4 $RW_p$ Algorithm and $RW_2$-based algorithms

### 4.1 Algorithms for Computing $RW_p$ Distances

We develop an efficient alternating optimization algorithm for computing $RW_p$ distance for general $p \geq 1$. The algorithm alternates between updating the transport plan $P$ and the translation vector $t$, forming a block-coordinate descent procedure that monotonically decreases the joint objective

$$\min_{t \in \mathbb{R}^d} \min_{P \in \Pi(a,b)} \sum_{i,j} P_{ij} \|x_i + t - y_j\|^p, \qquad \Pi(a,b) = \{P \geq 0 : P\mathbf{1} = a, \ P^\top \mathbf{1} = b\}.$$

Although the entire problem is non-convex, as mentioned in Subsection 3.2, each subproblem has convexity or linearity properties, allowing the overall method to remain computationally tractable.

**Overview of the alternating scheme.** The algorithm proceeds by fixing $t$ and solving for the optimal coupling $P$, then fixing $P$ and updating $t$ by minimizing the reduced convex objective. Each step is computationally simple: the $P$-update is a linear program, while the $t$-update is a smooth convex minimization.

**Updating the transport plan $P$.** When the translation $t$ is fixed, the problem reduces to a standard discrete optimal transport linear program with cost coefficients $C_{ij}(t) = \|x_i + t - y_j\|^p$. As $t$ changes across iterations, the feasible polytope $\Pi(a,b)$ remains fixed, and only the cost matrix is updated. As a result, a previously computed coupling $P^{(k)}$, together with its associated LP basis $B^{(k)}$, remains *primal feasible* for the next iteration. This enables warm-starting the LP using a dual simplex reinitialization step, avoiding the need to recompute the entire LP basis and significantly reducing computational cost.

**Updating the translation vector $t$.** For fixed $P$, the reduced objective

$$F_P(t) = \sum_{i,j} P_{ij} \|x_i + t - y_j\|^p$$

is convex in $t$. Instead of a single gradient step, we perform a short inner loop to approximately minimize $F_P(t)$, using gradient descent with an Armijo backtracking line search to ensure sufficient decrease and stability. This "inner solve" substantially improves descent efficiency, yet remains inexpensive because each step only involves evaluating weighted residuals of the form $x_i + t - y_j$.

**Geometric interpretation.** The feasible region $\Pi(a, b)$ is a fixed polytope in the space. For a given translation $t$, the matrix $C(t)$ defines an objective hyperplane whose slope depends on the direction and magnitude of $t$. Updating $t$ tilts this hyperplane, while the dual simplex step efficiently moves the solution to the new supporting face of the polytope. From this perspective, the alternating scheme repeatedly reshapes the geometry of the objective function and projects onto the polytope, tracing out a smooth descent path.

**Algorithm.** Algorithm 1 summarizes the procedure. We initialize $t$ using the means' difference, then alternate between warm-started LP solves and gradient-based updates of $t$ with Armijo backtracking. The objective decreases at every iteration.

---

**Algorithm 1** Alternating Optimization for $RW_p$ with Dual-Simplex-Acceleration

---

**Require:** Samples $\{x_i, a_i\}$, $\{y_j, b_j\}$, order $p \geq 1$, tolerances $\tau, \epsilon$, inner iteration cap $T_{\max}$
1: Initialize $t^{(0)} = \bar{\nu} - \bar{\mu}$
2: Solve OT with cost $C_{ij}^{(0)} = \|x_i + t^{(0)} - y_j\|^p$ to obtain $(P^{(0)}, B^{(0)})$
3: **repeat**
4: $\quad$ Update costs $C_{ij}^{(k)} = \|x_i + t^{(k)} - y_j\|^p$
5: $\quad$ Warm-start dual simplex to obtain $P^{(k+1)}$
6: $\quad$ $\tilde{t}^{(0)} \leftarrow t^{(k)}$
7: $\quad$ **for** $s = 0$ to $T_{\max} - 1$ **do**
8: $\quad\quad$ Compute gradient $g_s$ of $F_{P^{(k+1)}}$ at $\tilde{t}^{(s)}$
9: $\quad\quad$ Update $\tilde{t}^{(s+1)} = \tilde{t}^{(s)} - \alpha_s g_s$ using Armijo backtracking
10: $\quad\quad$ **if** $\|\tilde{t}^{(s+1)} - \tilde{t}^{(s)}\| \leq \epsilon \max\{1, \|\tilde{t}^{(s)}\|\}$ **then break**
11: $\quad$ $t^{(k+1)} = \tilde{t}^{(s+1)}$
12: $\quad$ $F^{(k+1)} = \sum_{i,j} P_{ij}^{(k+1)} \|x_i + t^{(k+1)} - y_j\|^p$
13: **until** $|F^{(k+1)} - F^{(k)}|/F^{(k)} < \tau$
14: **Output:** $(t^\star, P^\star, F^\star)$

---

**Convergence guarantee.** The following proposition formalizes the descent property of the algorithm. The full proof is provided in the Appendix A.5.

**Proposition 4.1** (Monotone descent and convergence). *Let $p \geq 1$ and assume $c(x, y) = \|x - y\|^p$ is differentiable for $p > 1$ (or admits a subgradient for $p = 1$). Then Algorithm 1 generates a non-increasing sequence of objective values $\{F^{(k)}\}$. Every accumulation point $(t^\star, P^\star)$ satisfies the first-order optimality conditions of the $RW_p$ problem. In addition, the warm-started dual simplex step yields locally linear convergence in the P-update when cost changes small, while the inner convex t-update (with Armijo backtracking) improves stability and accelerates overall descent.*

## 4.2 Applications of $RW_2$ Decomposition to Optimal Transport Computation

The decomposition results of Proposition 3.5 and Corollary 3.6 are useful for improving OT solvers. By separating the translational component from the intrinsic coupling structure, the new optimization has the same optimal coupling matrix while improving numerical stability and reducing computational errors. We describe its applications to both the LP-based algorithm and the Sinkhorn algorithm.

### 4.2.1 $RW_2$-LP Algorithm

For the LP-based OT problem, Corollary 3.6 implies that the Wasserstein cost can be separated into a translation term and a covariance term. When the objective coefficients $C_{ij} = \|x_i - y_j\|_2^2$ are extremely large, it might lead to ill-conditioned basis matrices. Translating the distributions can reduce the magnitude of the coefficients. Accordingly, one may translate the source distribution by $t^* = \bar{\nu} - \bar{\mu}$, compute the optimal transport plan between $(\mu + t^*, \nu)$, and then recover the full $W_2$ value. In practice, we introduce a threshold $M > 1$ to conditionally apply the translation operation only when it substantially reduces the maximum component, thereby avoiding unnecessary cost. More details can be found in Algorithm 2.

---

**Algorithm 2** The RW$_2$-LP Algorithm

---

**Require:** Empirical distributions $\mu = \sum_i a_i \delta_{x_i}$, $\nu = \sum_j b_j \delta_{y_j}$; threshold $M$
1: Compute $\bar{\mu}$, $\bar{\nu}$ and set $t^* = \bar{\nu} - \bar{\mu}$
2: Form costs $C_{ij} = \|x_i - y_j\|_2^2$ and $C'_{ij} = \|x_i + t^* - y_j\|_2^2$
3: **if** $M\|C'\|_\infty \leq \|C\|_\infty$ **then**                    ▷ Use mean alignment only when beneficial
4:     $C \leftarrow C'$
5: Solve $\min_{P \in \Pi(a,b)} \langle C, P \rangle$ via a linear programming OT solver
6: Output $W_2^2(\mu, \nu) = \|\bar{\mu} - \bar{\nu}\|_2^2 + \langle C, P^* \rangle$ and the optimal plan $P^*$

---

### 4.2.2 RW$_2$-**Sinkhorn Algorithm**

The same improvement can also be applied to the entropy-regularized OT problem. By performing the alignment conditionally controlled by threshold $M$, we can obtain the RW$_2$-Sinkhorn algorithm. In the following, we show that the convergence rate of the Sinkhorn algorithm actually remains the same, while the numerical stability is improved.

---

**Algorithm 3** RW$_2$-Sinkhorn Algorithm

---

**Require:** Measures $\mu = \sum_i a_i \delta_{x_i}$, $\nu = \sum_j b_j \delta_{y_j}$; regularizer $\lambda > 0$; tolerance $\varepsilon > 0$; constant $M$
1: Compute $\bar{\mu}$, $\bar{\nu}$ and $t^* = \bar{\nu} - \bar{\mu}$
2: Form costs $C$ and $C'$ as in Algorithm 2
3: **if** $M\|C'\|_\infty \leq \|C\|_\infty$ **then**
4:     $C \leftarrow C'$
5: Initialize $K = \exp(-C/\lambda)$, and $u = v = \mathbf{1}$
6: **repeat**
7:     $u \leftarrow a./(Kv), \quad v \leftarrow b./(K^\top u)$
8: **until** $\max\left(\|u \odot (Kv) - a\|, \|v \odot (K^\top u) - b\|\right) \leq \varepsilon$
9: $P^* = \mathrm{diag}(u)K\mathrm{diag}(v)$
10: Output $W_2^2(\mu, \nu) = \|t^*\|_2^2 + \langle C, P^* \rangle$

---

**Convergence rate.** Under translation $t$, the cost becomes $C'(t) = \|x_i + t - y_j\|^2$ and the Gibbs kernel of the Sinkhorn algorithm becomes $K' = \exp(-C'(t)/\lambda)$. We can prove that the contraction factor $\rho$ in Hilbert's projective metric is actually invariant under any translation $t$ (see Appendix A.6), where

$$\rho = \tanh\left(\frac{\Delta(K)}{4}\right), \qquad \Delta(K) = \frac{1}{\lambda} \max_{i,j,k,l} |C_{ik} + C_{jl} - C_{il} - C_{jk}|.$$

Thus, the aligned optimization has the same convergence rate as the original one.

**Numerical stability.** A principal source of numerical instability in Sinkhorn iterations is from underflow in the exponential kernel $K = \exp(-C/\lambda)$, particularly when $C_{ij}$ is large. The translation $t$ could alleviate this instability by conditionally reducing the magnitude of the cost coefficients,

$$C'_{ij} = \|x_i + t - y_j\|_2^2,$$

ensuring that the entries of $K' = \exp(-C'(t^*)/\lambda)$ remain well-scaled throughout the iterations. We may also measure this instability by defining the ill-condition of the matrix $K$ as

$$\kappa(K) = \prod_{i,j} K_{ij} = \exp\left(-\tfrac{1}{\lambda} \sum_{i,j} \|x_i + t - y_j\|_2^2\right).$$

By calculating the optimal condition of $\kappa(K)$, the maximizer occurs at $t = \bar{y} - \bar{x}$, which matches $t = \bar{\nu} - \bar{\mu}$ for empirical measures with uniform weights. As a result, the alignment can maximize $\kappa(K)$ and reduce kernel underflow.

**Computational complexity.** Altschuler et al. (2017) show that, for precision level $\tau_p$, time complexity of the Sinkhorn algorithm is $O\big(m^2\|C\|_\infty^3(\log m)\,\tau_p^{-3}\big)$, where $m = n_1 = n_2$ for simplicity. Since the alignment can reduce the value of $\|C\|_\infty$, our approach can also lower the complexity bound.

## 5 Experiments

We evaluate the proposed algorithms through three experiments. The first two experiments validate the numerical stability of the $RW_2$-based LP and Sinkhorn algorithms. The third experiment presents a real-world thunderstorm pattern retrieval task, illustrating the effectiveness of the general $RW_p$ distance in large-scale applications. All three experiments are conducted on a Linux workstation equipped with 64 CPU cores (Intel Core i7, 2.60 GHz), 16 GB RAM, and an NVIDIA RTX 3090 GPU.

### 5.1 Numerical Validation

We begin with validation tests to assess the numerical stability of the $RW_2$-based algorithms under varying dimensionalities and translation magnitudes. To provide a comprehensive evaluation of numerical stability, we also include normalization as a baseline in our experiments, as mentioned in Subsection 3.3. Here, normalization refers to scaling the cost matrix by dividing its maximum component prior to solving, and then rescaling the solution by the same factor.

#### 5.1.1 Validation of the $RW_2$-LP Algorithm

**Setup.** We consider two settings:

(1) *Same distribution with different translation*: The component of each sample in the source distribution $\mu$ is drawn from $\mathcal{N}(0,1)$, and the target distribution is constructed as $\nu = \mu + t$, where $t$ is a translation applied along the last coordinate and takes values in $\{1, 2, 4, 8, 16\}$. Each distribution size is 4,096 and we consider dimensions $d \in \{2, 10\}$ to represent both low- and high-dimensional settings.

(2) *Different distributions*: The component of each sample in the distribution $\mu$ is sampled from $\mathcal{N}(0,1)$. The component of each sample in $\nu$ is drawn from the Uniform distribution $\mathcal{U}[-1,1]$ first, and then the distribution $\nu$ is translated by $t = 1$ along the last coordinate. We consider $d \in \{2, 10\}$ with 2,048 samples for each distribution and vary the maximum iteration budget of the LP algorithm from $2^6$ to $2^{16}$.

We compare the $RW_2$-LP algorithm (Algorithm 2) with the standard LP algorithm. In addition, we also include the standard LP algorithm with normalization and the $RW_2$-LP algorithm with normalization as baselines, (translation is applied first, followed by normalization prior to OT solvers). Performance is evaluated in terms of the absolute error of $W_2^2(\mu, \nu)$ and the running time. The ground-truth value of $W_2^2(\mu, \nu)$ is given by $\|t\|_2^2$ in setting (1), and by a high-precision LP solution in setting (2). The LP solver is implemented using the `ot.emd2` function from the Python Optimal Transport (POT) library (Flamary et al., 2021), with the threshold parameter fixed at $M = 1$. Each experiment is repeated six times under the same settings.

**Results.** For the first setting, Figure 1(a) shows that the $RW_2$-LP formulation achieves substantially lower numerical errors than the standard LP solver, except when the translation magnitude equals 1. The error of the standard LP approach increases as the translation magnitude grows and becomes more pronounced in higher dimensions. In contrast, the $RW_2$-LP curves, both with and without normalization, almost overlap at the lower part of the plot, indicating consistently lower numerical errors. In addition, the normalization curves largely coincide with their corresponding original curves, suggesting that normalization alone does not significantly reduce the numerical errors. Figure 1(b) shows that the running time of $RW_2$-LP solvers remains comparable or slightly lower across all tested dimensions. Incorporating normalization may also lead to a slight reduction in running time.

For the second setting, Figure 2(a) shows that the $RW_2$-LP solvers, both with and without normalization, consistently attain lower error for all dimensions. Moreover, the normalization curves nearly coincide with their corresponding original curves, suggesting that normalization alone does not significantly reduce the

numerical errors. Figure 2(b) reports that running time of RW$_2$-LP solvers remains comparable across $d \in \{2, 10\}$. In summary, the $RW_2$ formulation improves the stability in the LP-based OT solver.

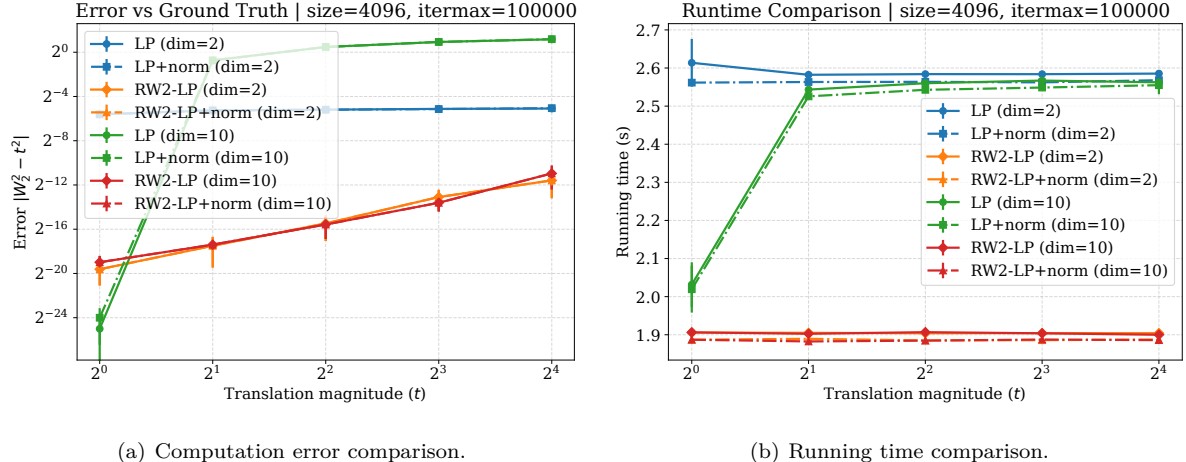

(a) Computation error comparison.  (b) Running time comparison.

Figure 1: Comparison of LP-based algorithms on Gaussian $\rightarrow$ Gaussian translation tasks. (a) The error of the standard LP approach increases as the translation magnitude grows and becomes more pronounced in higher dimensions. In contrast, the RW$_2$-LP curves across different dimensions almost completely overlap at the bottom of the plot, indicating consistently lower numerical errors. In addition, the RW$_2$-LP solvers, both with and without normalization, achieve substantially lower numerical errors than the standard LP solvers across all tested dimensions. (b) Running time of the RW$_2$-LP solvers remains comparable to the standard LP solvers across all tested dimensions. Incorporating normalization may also lead to a slight reduction in running time.

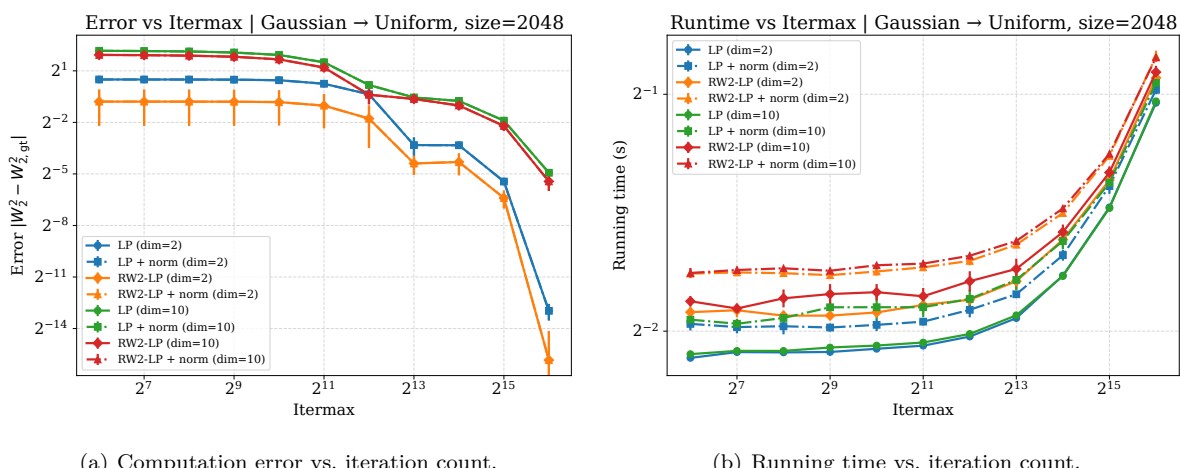

(a) Computation error vs. iteration count.  (b) Running time vs. iteration count.

Figure 2: LP algorithms comparison on Gaussian $\rightarrow$ Uniform tasks with limited iteration budgets. (a) The RW$_2$-LP solvers, both with and without normalization, achieve substantially lower error, especially under small budgets. The normalization curves nearly coincide with their corresponding original curves, suggesting that normalization alone does not reduce the numerical errors. (b) Running time of the RW$_2$-LP solvers remains comparable across $d \in \{2, 10\}$.

### 5.1.2 Validation of the $RW_2$-Sinkhorn Algorithm

**Setup.** We perform one validation test for the Sinkhorn algorithm under a configuration similar to setting (2) in the first experiment. The component of each sample in the source distribution $\mu$ is drawn from $\mathcal{N}(0,1)$, and the component of each sample in the target distribution $\nu$ is drawn from $\mathcal{U}[-1,1]$, then translated by $t \in \{1, 2, 4, 8, 16\}$. We test dimensions $d \in \{2, 10\}$ with 1,024 samples. We also test other pairs (Gaussian→Gaussian, Gaussian→Geometric, Gaussian→Poisson) with the same setting, and more results can be found in Appendix D.1. All these tests are repeated six times.

We compare the $RW_2$-Sinkhorn algorithm (Algorithm 3) with the standard Sinkhorn algorithm. In addition, we also include the $RW_2$-Sinkhorn algorithm with normalization and the standard Sinkhorn algorithm with normalization as baselines, (centering is applied first, followed by normalization prior to solving the optimal transport problem). All methods use `ot.sinkhorn2` from the POT library (Flamary et al., 2021) with regularization `reg` $= 10^{-5}$, a maximum of 1,000 iterations, and stopping threshold `stopThr` $= 10^{-5}$. The threshold parameter is fixed at $M = 1$.

**Results.** Figure 3(a) shows that the $RW_2$-Sinkhorn algorithms attain lower numerical errors, particularly as the magnitude of translation increases. The $RW_2$-Sinkhorn curves almost completely overlap at the bottom of the plot, indicating consistently low numerical errors across different dimensions. Moreover, the normalization curves lie below their corresponding original curves, suggesting that both centering and normalization can reduce errors, and centering provides a more substantial improvement than normalization. Figure 3(b) indicates that applying centering or normalization may introduce a slight increase in running time; however, the running time remains comparable across all configurations.

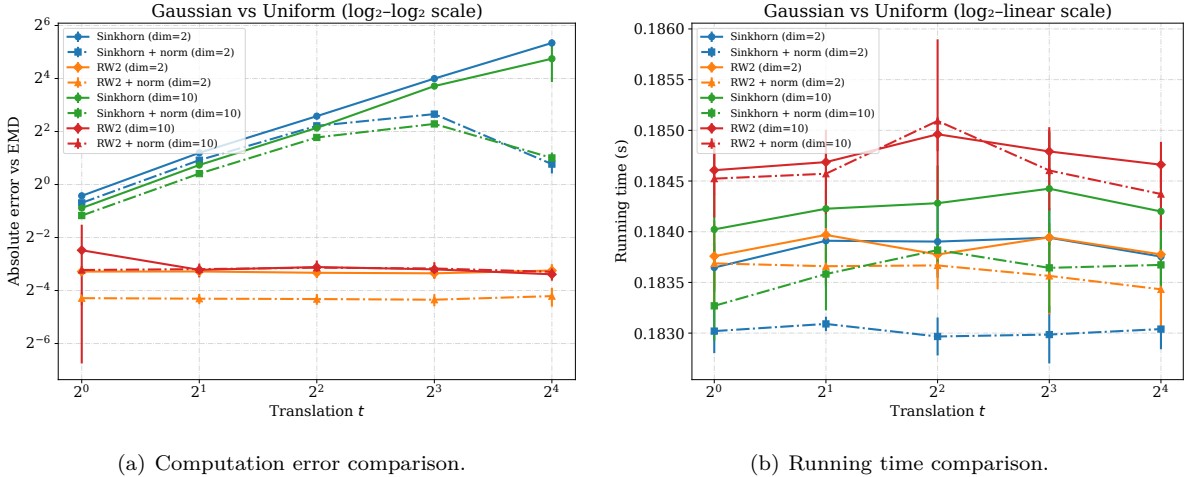

(a) Computation error comparison.                    (b) Running time comparison.

Figure 3: Sinkhorn algorithms comparison on Gaussian → Uniform tasks. (a) The $RW_2$-Sinkhorn algorithms, both with and without normalization, achieve lower numerical errors, particularly under large translations. The two $RW_2$-Sinkhorn curves almost completely overlap at the bottom of the plot, indicating consistently low numerical errors across dimensions. Moreover, while normalization curves also reduce errors relative to the original curves, centering provides more substantial improvements. (b) Applying centering or normalization may introduce a slight increase in running time; however, the running time remains comparable.

## 5.2 Thunderstorm Pattern Retrieval

In this subsection, we investigate the effectiveness of the proposed $RW_p$ distances ($p \in \{1, 2, 4\}$) for large-scale retrieval of real-world thunderstorm patterns. We firstly illustrate visual retrieval examples to show that $RW_p$ distances can capture structural similarities between storm events, and report the running time of each metric to assess their practical feasibility. Moreover, to fully evaluate retrieval performance, we also introduce a weakly supervised evaluation protocol to compare different metrics using precision–recall curve analysis.

Table 1: Running time for different metrics of retrieving the most similar thunderstorm snapshot from the dataset (32,073 images) given a reference snapshot. The centering time for $W_{1c}$ and $W_{4c}$ has been excluded.

| Metric | $\ell_2$ | $FSS$ | $W_2$ | $W_{1c}$ | $W_{4c}$ | $RW_1$ | $RW_2$ | $RW_4$ | $GW$ | $EGW$ |
|---|---|---|---|---|---|---|---|---|---|---|
| **Runtime(s)** | 11.16 | 11.83 | 50.87 | 374.95 | 404.49 | 1,060.15 | 48.28 | 803.21 | 11,903.57 | 12,919.69 |

Together, these experiments provide both qualitative and quantitative evidence to support the proposed metrics can be used to measure similarity for thunderstorm pattern retrieval.

### 5.2.1 Thunderstorm Pattern Retrieval with $RW_p$

Thunderstorm patterns are critical for airline and airport operations. Given reference thunderstorm events, it is useful to retrieve similar historical thunderstorm events in the database. We apply general $RW_p$ distances on the real-world thunderstorm dataset to show that general $RW_p$ can be used to retrieve similar thunderstorm patterns at large scale.

**Dataset and preprocessing.** Our data are collected radar images from MULTI-RADAR/MULTI-SENSOR SYSTEM (MRMS) (Zhang et al., 2016) focusing on a $300 \times 300$ $km^2$ rectangular area centered at the Dallas Fort Worth International Airport (DFW), where each pixel represents a $3 \times 3$ $km^2$ area. The snapshots are updated every 10 minutes, tracking from 2014 to 2022 between March and October, including around 32,073 images with thunderstorm patterns. Vertically Integrated Liquid Density (VIL density) and reflectivity are two common measurements for assessing thunderstorm intensity, with threshold values of $3kg \cdot m^{-3}$ and $35dBZ$, respectively (Dixon & Wiener, 1993; Matthews & Delaura, 2010; Wang et al., 2024). We use reflectivity as thunderstorm measurements and use $35dBZ$ as the intensity threshold to transform radar images to the corresponding *binary* matrices.

**Thunderstorm types:** We consider two types of thunderstorm events:

- **Snapshots**: a single radar image representing storm patterns;

- **Sequences**: a series of consecutive snapshots representing storm evolution in a short time.

In the main text we focus on snapshot retrieval; sequence-based results are provided in Appendix D.2.

**Snapshot retrieval.** Given a reference thunderstorm snapshot, we compute its distances to all snapshots in the dataset using $RW_p$ for $p \in \{1, 2, 4\}$. We compare these results with several baseline distances, including the $\ell_2$ distance, the Fractions Skill Score ($FSS$) (Roberts & Lean, 2008; Mittermaier, 2021), the Wasserstein distance $W_2$, and the centered Wasserstein distances $W_{1c}$ and $W_{4c}$, where $W_{pc}$ denotes the Wasserstein distance $W_p$ computed between centered distributions. In addition, we also include the Gromov–Wasserstein distance ($GW$) and the entropy-regularized Gromov–Wasserstein distance ($EGW$) as baseline methods, since they can also capture structural similarities between distributions.

For each distance, we retrieve the top-5 most similar thunderstorm snapshots. To increase the diversity of retrieval results, temporally redundant matches (within a 24-hour window) are removed, and only the closest match is retained within each window. Wasserstein distances are computed using the `emd2` function, while Gromov–Wasserstein ($GW$) and entropic Gromov–Wasserstein ($EGW$) distances are computed using the `gromov_gromov_wasserstein2` and `entropic_gromov_wasserstein2` functions from the POT library (Flamary et al., 2021). For $EGW$, we employ entropic regularization with $\varepsilon = 5 \times 10^{-2}$, the squared loss function, and a projected gradient descent (PGD) solver with a maximum of 200 iterations and a tolerance of $10^{-7}$. For $GW$, we use the same squared loss function, with a maximum of 200 iterations and a tolerance of $10^{-7}$. Pairwise cost matrices are constructed using Euclidean distances and are normalized to improve numerical stability. Figure 4 illustrates one example of snapshot retrieval results.

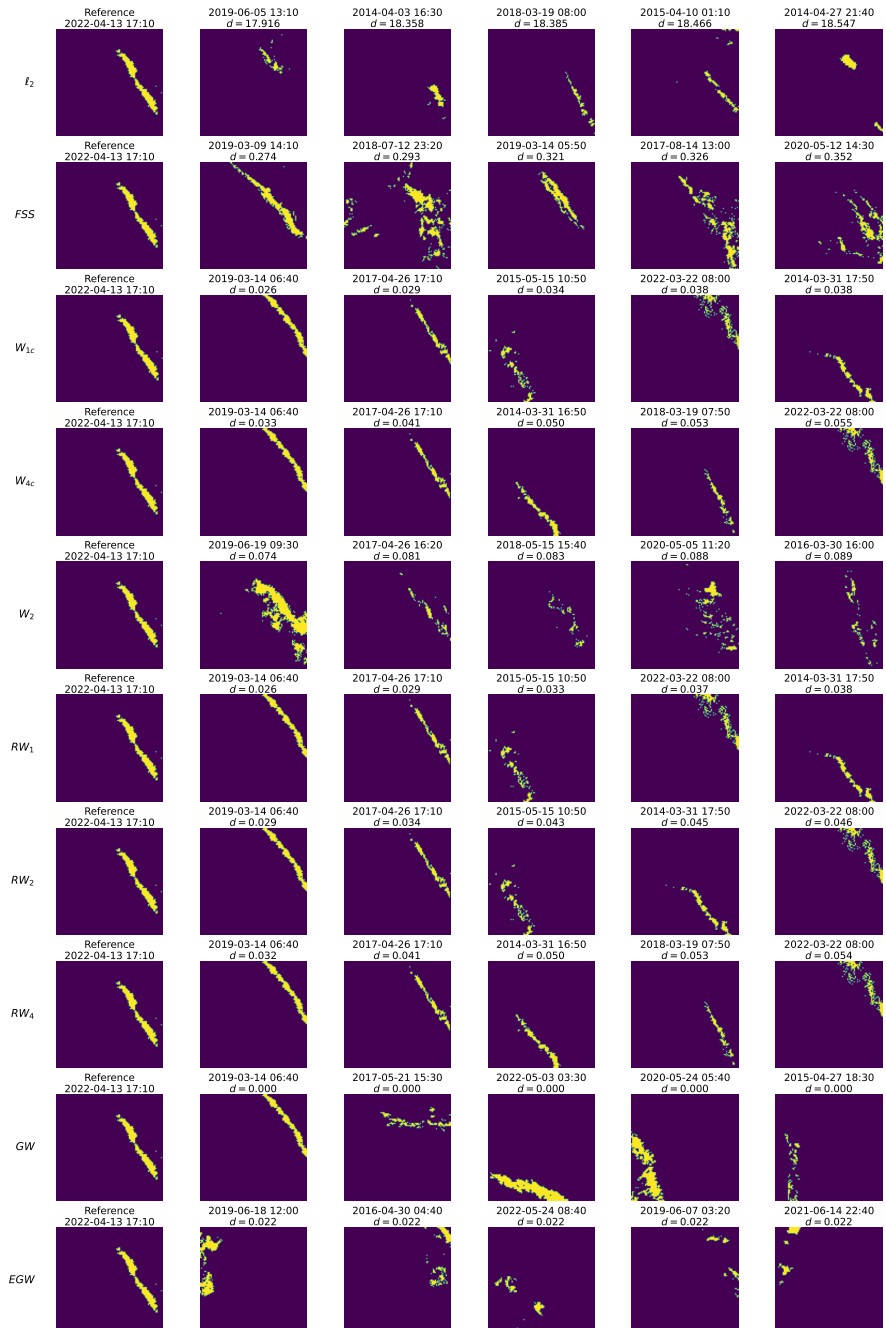

Figure 4: Top-5 retrieval results for different metrics given the same reference snapshot (leftmost column). Redundant retrievals that are temporally similar (within a 24-hour window) have been removed, and only the closest match within each window is retained. Rows correspond to $\ell_2$, $FSS$, $W_{1c}$, $W_{4c}$, $W_2$, $RW_p$ for $p \in \{1, 2, 4\}$, $GW$, and $EGW$. The value $d$ above each retrieved example denotes the corresponding distance between the reference and retrieved distributions. Except for $\ell_2$ and $FSS$, all metrics are computed on normalized coordinates to ensure numerical stability. Overall, $RW_p$ yields more matches in both shape and orientation.

**Results and analysis.** Table 1 summarizes the running time of each metric. The $\ell_2$ and $FSS$ distances are the fastest to compute. In contrast, $W_2$ requires substantially more time, reflecting the computational cost of solving the full optimal transport problem. The $RW_2$ formulation achieves a slightly lower running time

due to its decomposition property. Meanwhile, $W_{1c}$, $W_{4c}$, $RW_1$, and $RW_4$ incur additional computational overhead. $GW$ and $EGW$ distances exhibit the high running times (approximately 3 hours), as they rely on high dimensional cost matrices constructed from pairwise distances and involve solving a highly nonconvex optimization problem. Overall, these results indicate that the proposed method remains computationally feasible for large-scale applications.

Figure 4 presents the top-5 retrieval results obtained using different metrics for the same reference snapshot. The storms retrieved under the $\ell_2$ or $FSS$ distance are sparsely distributed and poorly aligned with the reference in both shape and orientation, indicating a limited ability to capture structural similarity. Retrievals based on $GW$ and $EGW$ distance capture certain aspects of structural similarity, such as the presence of two internal clusters shared by both the reference and retrieved storms; however, they may fail to preserve the correct orientations. $W_{1c}$ and $W_{4c}$, yield retrieval results that are almost the same as those obtained using $RW_1$ and $RW_4$. The main difference lies in the computed distances, where $RW_1$ and $RW_4$ are slightly smaller in some retrieval cases. This observation suggests that aligning distributions by their means does not always provide the optimal translation, as discussed in Subsection 3.3 and Appendix B.3. The classical Wasserstein distance $W_2$ improves retrieval quality by matching storms with more coherent mass distributions and orientations. Nevertheless, noticeable deformation and dispersion remain, reflecting its dependence on absolute spatial locations. In contrast, $RW_p$ distance retrieves thunderstorm events that closely match the reference in both shape and orientation for all tested orders $p \in \{1, 2, 4\}$. In particular, $RW_1$ is more tolerant to outliers and local noise, whereas increasing $p$ to 2 and 4 imposes stronger penalties on large transport, resulting in slightly higher distance values while preserving overall alignment. As illustrated by the sixth- to eighth-ranked retrievals in Figure 4, the thunderstorm event on 2022-03-22 exhibits a sparser spatial structure with more outliers than the event on 2014-03-31. Consequently, the latter achieves a smaller distance under the $RW_1$ distance, while the ranking is reversed under $RW_2$ and $RW_4$. This example highlights that $RW_1$ is more tolerant to outliers and local noise, whereas larger values of $p$ increasingly emphasize global shape similarity.

### 5.2.2 Quantitative Evaluation for Thunderstorm Pattern Retrieval

We provide precision–recall (PR) curves to quantitatively evaluate the retrieval performance. As there is currently no publicly available thunderstorm dataset with human-annotated labels that explicitly reflect structural similarity across different storm patterns, we adopt a weakly supervised evaluation protocol based on temporal consistency.

**Weakly supervised evaluation.** To enable quantitative evaluation, we adopt a weak annotation that leverages the natural temporal structure of the data. Specifically, given a temporal threshold $T_0$, for a reference snapshot at time $t$, we assign binary labels by categorizing snapshots within the temporal window $[t−T_0, t+T_0]$ as *similar*, and those outside this window as *dissimilar*. This protocol yields a temporally annotated dataset for retrieval evaluation. This labeling assumption is reasonable because thunderstorm systems typically evolve continuously and consistently in a short time, making temporally adjacent observations more likely to share similar spatial structures. Nevertheless, we also acknowledge that this constitutes a weak supervision strategy, since temporal proximity does not always imply structural similarity and may fail to capture certain meteorological dynamics.

**Translation enhancement and $\lambda_m$ tuning.** The proposed $RW_p$ distances primarily capture morphological similarity rather than spatial similarity and are inherently translation-invariant. While this property improves robustness to spatial alignment, it does not explicitly account for the physical displacement of storm systems between sequential radar snapshots. When the temporal gap between observations is relatively large (e.g., a radar refresh interval is 10 minutes), such displacement becomes a non-negligible factor in this supervised evaluation. To address this limitation, we also incorporate an additional translation regularization term weighted by a coefficient $\lambda_m$, which balances structural similarity and spatial displacement. The optimal value of $\lambda_m$ is selected via a linear search over the interval $[0.2, 1.6]$, and we empirically find that $\lambda_m = 0.6$ yields the best retrieval performance (see Figure 5(b)).

**Setup.** We compare the $\ell_2$ distance, the Wasserstein distance $W_2$, and the proposed relative Wasserstein distances $RW_p$ for $p \in \{1, 2, 4\}$, together with their translation-regularized variants denoted by $RW_p + \lambda_m$mean. Due to the high computational cost of retrieval, we did not include *GW* and *EGW* distances as baselines. Under the weakly supervised evaluation protocol, the temporal threshold is set to $T_0 = 35$ minutes, and the translation regularization coefficient is fixed to $\lambda_m = 0.6$, as determined by the tuning procedure described above. For each reference snapshot, retrieval performance is evaluated using top-3 ranking. Quantitative performance is assessed using precision–recall (PR) curves and related ranking metrics, aggregated over multiple queries.

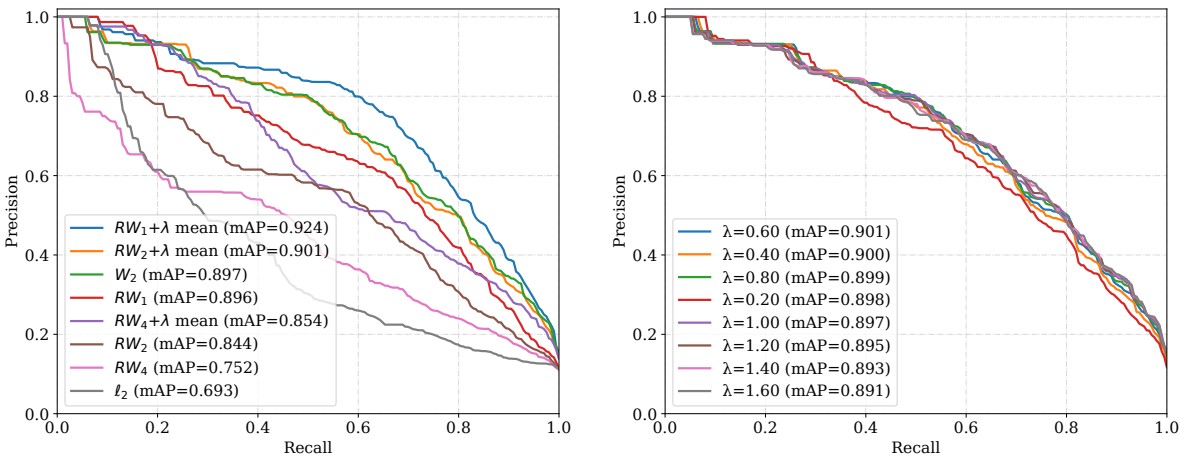

(a) Precision–Recall comparison of retrieval metrics.  (b) Precision–Recall comparison for regularizer $\lambda$ tuning.

Figure 5: Precision–recall (PR) curve comparison for thunderstorm pattern retrieval and translation regularization tuning. (a) Retrieval performance across different metrics, including the $\ell_2$ distance, the Wasserstein distance $W_2$, and the proposed relative Wasserstein distances $RW_p$ ($p \in \{1, 2, 4\}$), together with their translation-regularized variants $RW_p + \lambda_m$mean (with $\lambda_m = 0.6$). The regularized relative Wasserstein metrics achieve higher precision across recall levels, with $RW_1 + \lambda_m$mean attaining the best overall performance. (b) PR curves for tuning the translation regularization coefficient $\lambda_m$. A linear search over $\lambda_m \in [0.2, 1.6]$ indicates that $\lambda_m = 0.6$ yields the highest mean average precision, highlighting the benefit of jointly modeling structural similarity and storm displacement.

**Results and analysis.** Figure 5(a) presents the precision–recall (PR) curves for thunderstorm pattern retrieval under the weak temporal supervision protocol. Among all compared methods, the regularized metric $RW_1 + \lambda_m$mean achieves the best overall performance in terms of mean average precision (mAP), suggesting that lower-order relative transport cost provides more robust alignment for thunderstorm pattern retrieval compared to higher-order variants. Figure 5(b) illustrates the effect of different translation regularization parameters. A systematic search over $\lambda_m$ reveals that moderate regularization yields the best retrieval accuracy, with $\lambda_m = 0.6$ producing the highest mean average precision. This result highlights the importance of jointly modeling morphological similarity and spatial displacement in thunderstorm pattern retrieval.

## 6 Conclusions

In this paper, we introduce a novel family of distances, relative translation invariant Wasserstein distances $(RW_p)$, for measuring the similarity between probability distributions. Extended from the classical optimal transport framework, we show that $RW_p$ defines a proper metric on the quotient space $\mathcal{P}_p(\mathbb{R}^d)/\sim_T$ and is invariant under relative translations. In the special case $p = 2$, the proposed distance exhibits additional structure, including a decomposition of the optimal transport formulation and translation-invariant optimal coupling matrix. We further develop algorithms for computing general $RW_p$ distances, and $RW_2$-based algorithms for both LP-based and Sinkhorn OT solvers to improve numerical stability and reduce computational

errors. Finally, we validate the proposed algorithms through three experiments, demonstrating that our proposed algorithms can reduce computational errors in both LP-based and Sinkhorn OT solvers and enable practical meteorological applications in large-scale real-world settings.

## 7 Acknowledgements

We would like to express our sincere gratitude to Timothy Niznik, Yuqiang Wang, Xufang Zheng, Deng Na, Hannah Smith and Ian Ayers at American Airlines for their support, helpful discussions, and valuable feedback for the thunderstorm project. We would also like to thank Matthias Steiner and James Pinto from the National Center for Atmospheric Research (NCAR), as well as Richard DeLaura from MIT Lincoln Laboratory, for their insightful guidance and expertise on weather-related aspects of this work. In addition, we gratefully acknowledge partial financial support for this research from the National Science Foundation (Award No. 2047390).

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

# Appendix

## A  Proofs

### A.1  Proposition 3.2

*Proof of Proposition 3.2.* Let $p \in [1, \infty)$ and $\mu, \nu \in \mathcal{P}_p(\mathbb{R}^d)$. Define $W_p$ as the Wasserstein distance with cost $\|x - y\|^p$, and set $F(t) := W_p(\mu + t, \nu)$.

For all $t \in \mathbb{R}^d$,

$$W_p(\mu, \mu + t) = \|t\|.$$

By the triangle inequality,

$$F(t) \ \geq \ \big| W_p(\mu + t, \mu) - W_p(\mu, \nu) \big| = \big| \|t\| - W_p(\mu, \nu) \big| = \max\{\|t\| - W_p(\mu, \nu), W_p(\mu, \nu) - \|t\|\}.$$

Hence if $\|t\| \geq 2 W_p(\mu, \nu)$, then

$$F(t) \ \geq \ \|t\| - W_p(\mu, \nu) \ \geq \ W_p(\mu, \nu) = F(0).$$

So no minimizer lies outside the ball

$$B := \{ t \in \mathbb{R}^d : \|t\| \leq 2 W_p(\mu, \nu) \}.$$

Since $F$ is lower semi-continuous in $t$ and $B$ is compact, $F$ attains its minimum on $B$. Therefore,

$$\mathrm{ROT}(\mu, \nu, p) \ = \ \min_{\|t\| \leq 2 W_p(\mu, \nu)} W_p(\mu + t, \nu).$$

$\square$

### A.2  Theorem 3.4

*Proof of Theorem 3.4.* Using the previous notations, we first verify that the translation relation $\sim_T$ is an equivalence relation on $\mathcal{P}_p(\mathbb{R}^d)$. It is reflexive, since any $\mu \in \mathcal{P}_p(\mathbb{R}^d)$ can be translated to itself by the zero vector; symmetric, since if $\mu$ can be translated to $\nu$, then $\nu$ can be translated back to $\mu$; and transitive, since if $\mu$ can be translated to $\nu$ and $\nu$ to $\eta$, then $\mu$ can also be translated to $\eta$.

Hence, by the properties of equivalence relations, the quotient set $\mathcal{P}_p(\mathbb{R}^d)/\sim_T$ is well defined. Let $[\mu]$ denote an element of this quotient space. Based on that, $W_p(\cdot, \cdot)$ is a true metric on $\mathcal{P}_p(\mathbb{R}^d)$ (Villani, 2003), it satisfies identity, positivity, symmetry, and the triangle inequality. We now show that $RW_p(\cdot, \cdot)$ also satisfies these axioms on $\mathcal{P}_p(\mathbb{R}^d)/\sim_T$.

For any $[\mu], [\nu], [\eta] \in \mathcal{P}_p(\mathbb{R}^d)/\sim_T$:

- Identity:

$$RW_p([\mu], [\mu]) = \min_{\mu', \mu'' \in [\mu]} W_p(\mu', \mu'') = W_p(\mu', \mu') = 0.$$

- Positivity:

$$RW_p([\mu], [\nu]) = \min_{\mu' \in [\mu], \nu' \in [\nu]} W_p(\mu', \nu') \geq 0.$$

- Symmetry:

$$RW_p([\mu], [\nu]) = \min_{\mu' \in [\mu], \nu' \in [\nu]} W_p(\mu', \nu') = \min_{\nu' \in [\nu], \mu' \in [\mu]} W_p(\nu', \mu') = RW_p([\nu], [\mu]).$$

- Triangle inequality:

  Fix $\epsilon > 0$. By definition of the minimum, there exist $\mu' \in [\mu], \nu' \in [\nu]$ such that

  $$W_p(\mu', \nu') \leq RW_p([\mu], [\nu]) + \epsilon,$$

  and $\nu'' \in [\nu], \eta' \in [\eta]$ such that

  $$W_p(\nu'', \eta') \leq RW_p([\nu], [\eta]) + \epsilon.$$

  Since $\nu' \sim_T \nu''$, there exists a translation $t \in \mathbb{R}^d$ with $\nu'' = \nu' - t$. By translation invariance of $W_p$, $(W_p(\mu + t, \nu + t) = W_p(\mu, \nu), \forall t \in \mathbb{R}^d, \forall \mu, \nu \in \mathcal{P}_p(\mathbb{R}^d))$, we have

  $$W_p(\nu'', \eta') = W_p(\nu' - t, \eta') = W_p(\nu' - t + t, \eta' + t) = W_p(\nu', \eta' + t).$$

  And the triangle inequality for $W_p$ gives

  $$W_p(\mu', \eta' + t) \leq W_p(\mu', \nu') + W_p(\nu', \eta' + t).$$

  Combining with the above bounds,

  $$\begin{aligned}
  RW_p([\mu], [\eta]) &\leq W_p(\mu', \eta' + t) \\
  &\leq W_p(\mu', \nu') + W_p(\nu', \eta' + t) \\
  &= W_p(\mu', \nu') + W_p(\nu'', \eta') \\
  &\leq RW_p([\mu], [\nu]) + RW_p([\nu], [\eta]) + 2\epsilon.
  \end{aligned}$$

  Since $\epsilon > 0$ was arbitrary, the inequality follows.

Therefore, $RW_p$ defines a metric on $\mathcal{P}_p(\mathbb{R}^d)/\sim_T$. $\qquad\square$

### A.3 Proof of Proposition 3.5

*Proof of Proposition 3.5.* We first establish the decomposition in the continuous setting and then verify the invariance of the optimal coupling matrix in the discrete case.

**Continuous case.** Consider the quadratic ROT problem

$$\mathrm{ROT}(\mu, \nu, 2) = \min_{t \in \mathbb{R}^d} \min_{\gamma \in \Pi(\mu+t, \nu)} \int_{\mathbb{R}^{2d}} \|x + t - y\|_2^2 \, d\gamma(x, y).$$

Expanding the square yields

$$\begin{aligned}
\int \|x + t - y\|_2^2 \, d\gamma &= \int \left( \|x - y\|_2^2 + \|t\|_2^2 + 2t \cdot (x - y) \right) d\gamma \\
&= \int \|x - y\|_2^2 \, d\gamma + \|t\|_2^2 + 2t \cdot \int (x - y) \, d\gamma.
\end{aligned} \tag{5}$$

For any $\gamma \in \Pi(\mu, \nu)$, the marginal conditions imply

$$\int x \, d\gamma = \bar{\mu}, \qquad \int y \, d\gamma = \bar{\nu}.$$

Thus, $\int (x - y) \, d\gamma = \bar{\mu} - \bar{\nu}$. Substituting into 5 gives

$$\int \|x + t - y\|_2^2 d\gamma = \int \|x - y\|_2^2 d\gamma + \|t\|_2^2 + 2t \cdot (\bar{\mu} - \bar{\nu}).$$

Thus,

$$\mathrm{ROT}(\mu, \nu, 2) = \mathrm{OT}(\mu, \nu, 2) + \min_{t \in \mathbb{R}^d} \left( \|t\|_2^2 + 2t \cdot (\bar{\mu} - \bar{\nu}) \right),$$

and the strictly convex term is minimized at $t^* = \bar{\nu} - \bar{\mu}$, yielding

$$\mathrm{ROT}(\mu, \nu, 2) = \mathrm{OT}(\mu, \nu, 2) - \|\bar{\mu} - \bar{\nu}\|_2^2.$$

**Discrete case (invariance of the optimal coupling matrix).** Let $\mu = \sum_{i=1}^{n_1} a_i \delta_{x_i}$ and $\nu = \sum_{j=1}^{n_2} b_j \delta_{y_j}$, and let $P \in \Pi(a,b)$ be a coupling matrix. For fixed $t$, the discrete ROT objective is

$$\sum_{i,j} P_{ij} \|x_i + t - y_j\|_2^2.$$

Expanding the square gives

$$\sum_{i,j} P_{ij} \|x_i + t - y_j\|_2^2 = \sum_{i,j} P_{ij} \|x_i - y_j\|_2^2 + \|t\|_2^2 \sum_{i,j} P_{ij} + 2t \cdot \sum_{i,j} P_{ij}(x_i - y_j). \tag{6}$$

Using the marginal constraints,

$$\sum_{i,j} P_{ij} = 1, \qquad \sum_{i,j} P_{ij} x_i = \bar{\mu}, \qquad \sum_{i,j} P_{ij} y_j = \bar{\nu},$$

hence $\sum_{i,j} P_{ij}(x_i - y_j) = \bar{\mu} - \bar{\nu}$. Substituting into 6, we obtain

$$\sum_{i,j} P_{ij} \|x_i + t - y_j\|_2^2 = \sum_{i,j} P_{ij} \|x_i - y_j\|_2^2 + \|t\|_2^2 + 2t \cdot (\bar{\mu} - \bar{\nu}),$$

where the additional terms depend only on $t$ and not on $P$. Thus, for any fixed $t$, the minimizers over $P \in \Pi(a,b)$ of

$$P \mapsto \sum_{i,j} P_{ij} \|x_i + t - y_j\|_2^2$$

coincide exactly with those of the original quadratic OT objective

$$P \mapsto \sum_{i,j} P_{ij} \|x_i - y_j\|_2^2.$$

Hence, the optimal coupling is invariant under any relative translation.

Combining the continuous decomposition with the discrete invariance establishes the proposition. $\qquad\square$

### A.4 Analytical Characterization of the ROT Problem in One Dimension

In one dimension, the ROT problem admits an analytical characterization through the quantile formulation of optimal transport, which follows from the monotone rearrangement structure of optimal transport between distributions. Let $\mu, \nu \in \mathcal{P}_p(\mathbb{R})$ with $p \geq 1$, and denote by $F_\mu$ and $F_\nu$ their cumulative distribution functions and by $F_\mu^{-1}$ and $F_\nu^{-1}$ their corresponding quantile functions. The $p$-Wasserstein distance between $\mu$ and $\nu$ can be expressed as

$$W_p^p(\mu, \nu) = \int_0^1 \left|F_\mu^{-1}(s) - F_\nu^{-1}(s)\right|^p ds, \tag{7}$$

which follows from the monotone rearrangement structure of optimal transport in one dimension (Villani, 2009; Peyré & Cuturi, 2019).

Using this formulation, the relative optimal transport (ROT) problem

$$\mathrm{ROT}(\mu, \nu, p) = \inf_{t \in \mathbb{R}} W_p^p(\mu + t, \nu)$$

can be written explicitly as

$$\mathrm{ROT}(\mu, \nu, p) = \inf_{t \in \mathbb{R}} \int_0^1 \left|F_\mu^{-1}(s) + t - F_\nu^{-1}(s)\right|^p ds. \tag{8}$$

Define

$$\Delta(s) := F_\nu^{-1}(s) - F_\mu^{-1}(s), \quad s \in (0,1).$$

Then the optimization problem becomes

$$\text{ROT}(\mu, \nu, p) = \inf_{t \in \mathbb{R}} \int_0^1 |t - \Delta(s)|^p ds. \tag{9}$$

The objective function

$$\phi(t) := \int_0^1 |t - \Delta(s)|^p ds$$

is convex with respect to $t$, and therefore the ROT problem in one dimension reduces to a one-dimensional convex optimization problem.

**Proposition A.1.** *Let $\mu, \nu \in \mathcal{P}_p(\mathbb{R})$. The optimal translation $t^\star$ solving the ROT problem satisfies*

$$t^\star \in \arg\min_{t \in \mathbb{R}} \int_0^1 |t - \Delta(s)|^p ds.$$

*In particular:*

- *if $p = 1$, any median of $\Delta(s)$ is an optimal translation;*

- *if $p > 1$, the minimizer is unique and satisfies*

$$\int_0^1 |t^\star - \Delta(s)|^{p-2}(t^\star - \Delta(s))\, ds = 0.$$

In the quadratic case ($p = 2$), the optimal translation admits a closed-form expression:

$$t^\star = \int_0^1 \Delta(s)\, ds = \int_0^1 \left( F_\nu^{-1}(s) - F_\mu^{-1}(s) \right) ds = \bar{\nu} - \bar{\mu}.$$

Therefore, when $p = 2$, the optimal translation coincides with the difference of the means, which leads to the decomposition of the quadratic Wasserstein distance discussed in Subsection 3.3.

### A.5 Proof of Proposition 4.1

*Proof.* Recall that the objective of the $RW_p$ problem is

$$F(t, P) = \sum_{i,j} P_{ij} \|x_i + t - y_j\|^p,$$

with $p \geq 1$. The feasible set $\Pi(a, b)$ is convex and compact, and for any fixed coupling $P$, the map $t \mapsto F(t, P)$ is convex (strictly convex when $p > 1$).

**Step 1: Monotone descent.** Each iteration consists of two substeps.

*(a) P-update.* For fixed $t^{(k)}$, the coupling is updated by solving the linear program

$$P^{(k+1)} = \arg\min_{P \in \Pi(a,b)} F(t^{(k)}, P),$$

which gives

$$F(t^{(k)}, P^{(k+1)}) \leq F(t^{(k)}, P^{(k)}).$$

The warm-started dual simplex step used in Algorithm 1 preserves this monotone decrease.

*(b) t-update.* For fixed $P^{(k+1)}$, the algorithm performs an inner gradient-based minimization of the convex function

$$t \mapsto F(t, P^{(k+1)}),$$

using Armijo backtracking to choose the step size. Therefore, each inner step satisfies the sufficient decrease condition

$$F(t^{(k+1)}, P^{(k+1)}) \leq F(t^{(k)}, P^{(k+1)}).$$

Combining the two substeps yields the global descent property

$$F(t^{(k+1)}, P^{(k+1)}) \leq F(t^{(k)}, P^{(k)}), \qquad \forall k \geq 0,$$

so $\{F^{(k)}\}$ is a non-increasing sequence bounded below by 0, and therefore convergent.

**Step 2: Existence of accumulation points.** Because $\Pi(a,b)$ is compact and $F(\cdot, P)$ is coercive in $t$ for each $P$, the sequence $\{t^{(k)}\}$ remains bounded. Thus, the sequence $\{(t^{(k)}, P^{(k)})\}$ admits at least one accumulation point $(t^\star, P^\star)$.

**Step 3: Stationarity of accumulation points.** For each $k$, we have the optimality relation

$$P^{(k+1)} = \arg \min_{P \in \Pi(a,b)} F(t^{(k)}, P),$$

and the Armijo-based inner loop ensures that $t^{(k+1)}$ satisfies a first-order decrease condition for the convex problem $\min_t F(t, P^{(k+1)})$. Passing to the limit along any convergent subsequence and using continuity of $F$ and of its gradient (or subgradient) in $t$, we obtain

$$0 \in \partial_t F(t^\star, P^\star), \qquad P^\star \in \arg \min_{P \in \Pi(a,b)} F(t^\star, P).$$

Hence, $(t^\star, P^\star)$ satisfies the first-order optimality conditions of the $RW_p$ problem.

**Step 4: Conclusion.** The alternating scheme produces a monotone sequence of objective values converging to $F^\star$, and every accumulation point of the iterates is a stationary point of the non-convex $RW_p$ problem. The dual simplex re-initialization ensures locally linear progress in the $P$-update when the cost perturbation is small, while the Armijo-controlled inner $t$-update guarantees stable and accelerated descent.

$\square$

## A.6   Invariance of the Sinkhorn Convergence Rate under Translation

The following proposition formalizes the invariance of the Sinkhorn convergence rate under translation. In particular, it establishes that translating the input distributions does not affect the contraction constant of the Sinkhorn operator in Hilbert's projective metric, even if it improves numerical conditioning.

**Proposition A.2** (Translation invariance of the Hilbert–metric contraction). *Let $C_{ij} = \|x_i - y_j\|_2^2$ denote the quadratic cost matrix, and let*

$$K = \exp\left(-\frac{C}{\lambda}\right), \qquad \lambda > 0,$$

*be the associated Gibbs kernel used in the Sinkhorn algorithm. For any translation vector $t \in \mathbb{R}^d$, define the translated cost*

$$C'_{ij} = \|x_i + t - y_j\|_2^2, \qquad K' = \exp\left(-\frac{C'}{\lambda}\right).$$

*Then the projective diameter of $K$,*

$$\Delta(K) = \frac{1}{\lambda} \sup_{i,j,k,l} |C_{ik} + C_{jl} - C_{il} - C_{jk}|,$$

*satisfies $\Delta(K') = \Delta(K)$. Consequently, the Hilbert metric contraction factor*

$$\rho = \tanh\left(\frac{\Delta(K)}{4}\right),$$

*and hence, the geometric convergence rate of the Sinkhorn iterations is invariant under translation of the input distributions.*

*Proof.* We begin by expanding the translated cost function:

$$C'_{ij} = \|x_i + t - y_j\|_2^2 = \|x_i - y_j\|_2^2 + 2t \cdot (x_i - y_j) + \|t\|_2^2.$$

Substituting into the expression defining the projective diameter yields

$$C'_{ik} + C'_{jl} - C'_{il} - C'_{jk} = (C_{ik} + 2t\cdot(x_i - y_k) + \|t\|_2^2) + (C_{jl} + 2t\cdot(x_j - y_l) + \|t\|_2^2)$$
$$- (C_{il} + 2t\cdot(x_i - y_l) + \|t\|_2^2) - (C_{jk} + 2t\cdot(x_j - y_k) + \|t\|_2^2).$$

The constant terms $\|t\|_2^2$ cancel since they appear twice with positive and twice with negative sign. The linear terms in $t$ also cancel, as

$$(x_i - y_k) + (x_j - y_l) - (x_i - y_l) - (x_j - y_k) = 0.$$

Therefore,

$$C'_{ik} + C'_{jl} - C'_{il} - C'_{jk} = C_{ik} + C_{jl} - C_{il} - C_{jk},$$

implying that $\Delta(K') = \Delta(K)$.

As a consequence, the contraction factor $\rho = \tanh(\Delta(K)/4)$ and the corresponding geometric convergence rate of the Sinkhorn algorithm remains unchanged under any translation $t \in \mathbb{R}^d$. Although translation affects the numerical scaling of the kernel entries and thereby their conditioning, it leaves the convergence rate invariant. □

## B    Examples of ROT Problems

In this section, we present three examples to show that finding the exact solution of the ROT problem for $p \geq 1$ is challenging in general. The first two examples demonstrate the non-convexity of the ROT formulation in a two-dimensional setting, and the third example shows that the optimal solution $t$ is not necessarily identical to the difference between the means, when the order $p \neq 2$.

Throughout this section, the underlying cost function is assumed to be

$$c(x, y) = \|x - y\|_p^p.$$

### B.1    Non-convexity with respect to the translation variable $t$

Consider a two-dimensional setting where the source and target distributions $\mu$ and $\nu$ are supported on

$$\{x_i = (\cos\frac{2i\pi}{3}, \sin\frac{2i\pi}{3}),\ i = 1, 2, 3\}, \qquad \{y_j = (-\cos\frac{2j\pi}{3}, -\sin\frac{2j\pi}{3}),\ j = 1, 2, 3\},$$

each with equal masses. These configurations are illustrated in Figure 6(a).

We focus on the effect of the translation variable $t$ on the objective function and the transport plan $P$ attains its minimal value for each fixed $t$. In other words, we study the function

$$t \ \mapsto\ \min_P \sum_{i,j} P_{ij} \|x_i + t - y_j\|_p^p.$$

Figures 6(b)–(c) show the corresponding contour and surface plots for $p = 1$, both clearly indicating that this function is non-convex with respect to $t$.

Using the same $\mu$ and $\nu$, we also examine other exponents $p \in \{1.2, 4, 10\}$. The corresponding contour and surface plots are shown in Figure 7, indicating that non-convexity in $t$ persists for a range of $p$ values.

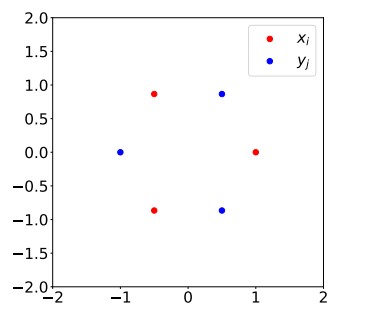 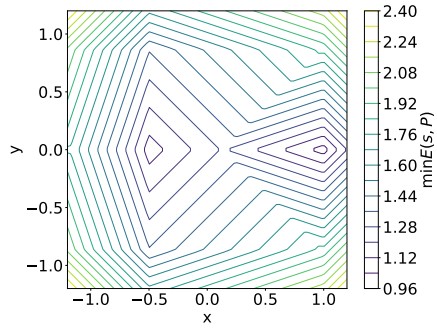 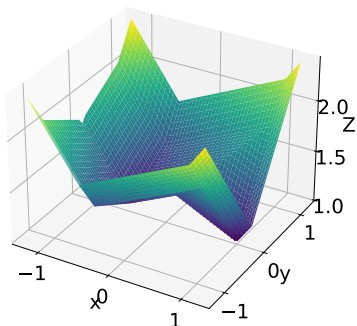

(a) Distributions $\mu$ and $\nu$.

(b) Contour plot of the function $\min_P \sum_{i,j} P_{ij} \|x_i + t - y_j\|$ with respect to $t$.

(c) Surface plot of the function $\min_P \sum_{i,j} P_{ij} \|x_i + t - y_j\|$ with respect to the translation vector $t = (x,y)$.

Figure 6: Contour and surface plots of $\min_P \sum_{i,j} P_{ij} \|x_i + t - y_j\|$ showing non-convexity in $t$.

## B.2 Non-convexity with respect to the coupling variable $P$

Next, using the same source and target distributions as in the first example and setting $p = 1$, we examine the effect of the variable $P$ on the objective function via fixing the translation vector $t$ to be its minimal value for each fixed $P$. In other words, we consider the function

$$F_1(P) = \min_t \sum_{i,j} P_{ij} \|x_i + t - y_j\|,$$

and show that $F_1(P)$ is non-convex with respect to the variable $P$.

Since the dimension of $P$ is high, plotting the contour of $F_1(P)$ directly is not possible. Instead, we demonstrate non-convexity by exhibiting two transport plans $P_1$ and $P_2$ such that the interpolated function value is strictly smaller than the function value at the interpolated transport plan, which violates the convexity property. Therefore, the function $F_1(P)$ is non-convex.

Consider two feasible transport plans:

$$P_1 = \tfrac{1}{3} \begin{bmatrix} 1 & 0 & 0 \\ 0 & 0 & 1 \\ 0 & 1 & 0 \end{bmatrix}, \qquad P_2 = \tfrac{1}{3} \begin{bmatrix} 0 & 1 & 0 \\ 0 & 0 & 1 \\ 1 & 0 & 0 \end{bmatrix}.$$

The optimal translations are $t_{P_1}^* = (1,0)$ and $t_{P_2}^* = (-\tfrac{1}{2},0)$, therefore, $F_1(P_1) = 1$ and $F_1(P_2) = \tfrac{1}{2} + \tfrac{\sqrt{3}}{3}$. However, for their midpoint, $\tfrac{1}{2}(P_1 + P_2)$, we obtain

$$F_1\left(\tfrac{1}{2}(P_1 + P_2)\right) = 1 + \tfrac{\sqrt{3}}{6} > \tfrac{1}{2}(F_1(P_1) + F_1(P_2)),$$

which shows that $F_1(P)$ is non-convex in $P$.

In summary, the above examples show that the convexity of the ROT problem cannot be guaranteed in general, especially in high-dimensional or non-quadratic cases.

## B.3 Optimal translation is not necessarily identical to the difference between the means

We now show that for $p \neq 2$, the optimal translation minimizing

$$\min_t \sum_{i,j} P_{ij} \|x_i + t - y_j\|_p^p$$

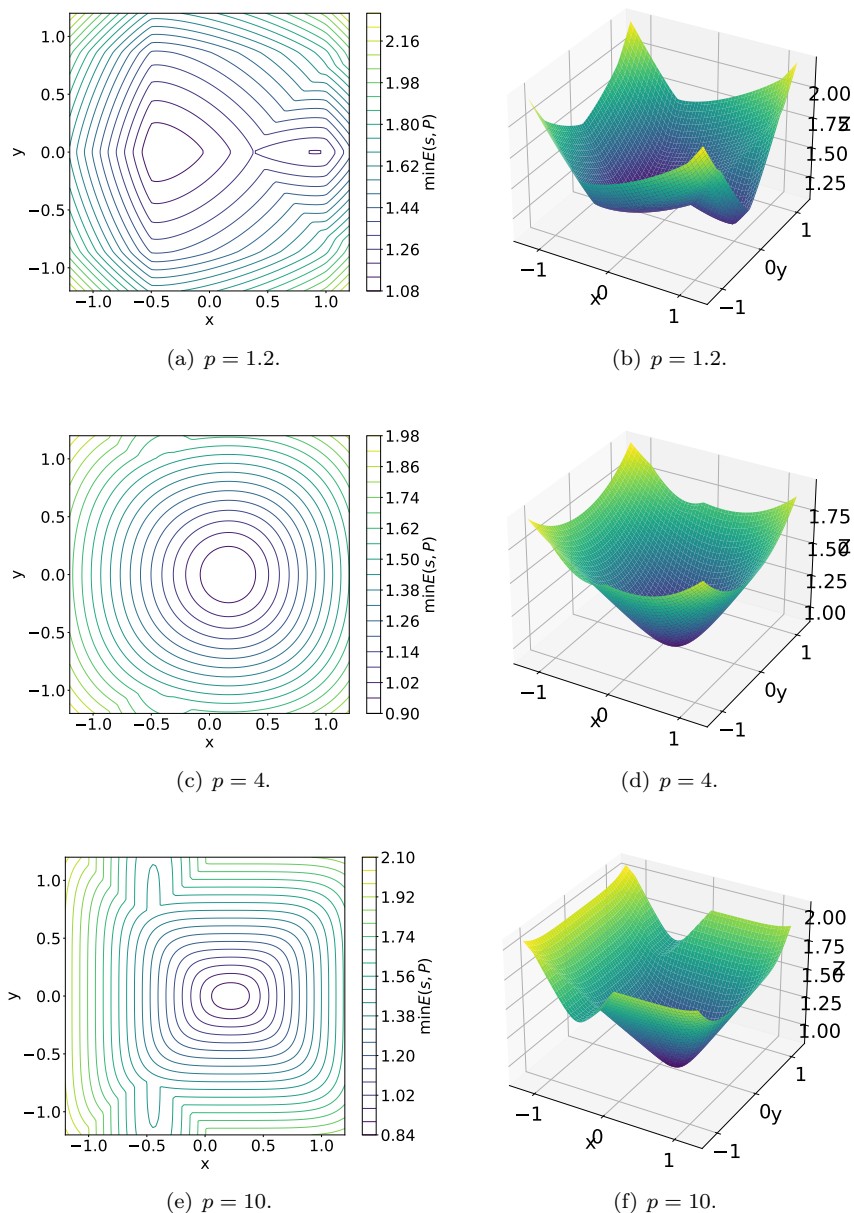

Figure 7: Contour and surface plots of the function $\min_P \sum_{i,j} P_{ij} \|x_i + t - y_j\|_p^p$ showing non-convexity in $t$ for $p \in \{1.2, 4, 10\}$.

does not necessarily coincide with the difference between the mean vectors of the two distributions.

Consider the following two-dimensional example:

$$\mu := \{x_1 = (3,0),\ x_2 = (0,0),\ x_3 = (0,3)\}, \qquad \nu := \{y_1 = (-3,0),\ y_2 = (0,0),\ y_3 = (0,-3)\},$$

where all points have equal mass.

For $p = 1$, the mean vectors are $\bar{\mu} = (0,0)$ and $\bar{\nu} = (0,0)$. Using their difference as the translation gives

$$W_1(\mu, \nu) = \tfrac{1}{3}(3 + 3 + 6) = 4.$$

However, when translating the source distribution by $t_0 = (-3, -3)$, we obtain

$$W_1(\mu + t_0, \nu) = \tfrac{1}{3}(3 + 3) = 2 < 4.$$

Therefore, for $p \neq 2$, optimal translation is not necessarily identical to the difference between the means of two distributions.

## C   Rotation Equivalence

In addition to the translation relation, an extension of the ROT framework is to consider the *rotation* relation on the space of probability distributions. This perspective allows us to explore the geometric structure of distributions up to rigid-body rotations or reflections, and to study the computational behavior of the resulting optimization problem. This rotation concept is closely related to the Procrustes–Wasserstein distance and sliced Gromov–Wasserstein methods. For completeness, we also present the results and proofs for this distance; see (Zhang et al., 2017; Vayer et al., 2022; Adamo et al., 2025) for further detailed discussion.

### C.1   Rotation equivalence and induced quotient metric

Let $O(n)$ denote the orthogonal group of $\mathbb{R}^d$, consisting of all rotations and reflections. We define an equivalence relation $\sim_R$ on $\mathcal{P}_p(\mathbb{R}^d)$ as follows:

$$\mu \sim_R \nu \quad \Longleftrightarrow \quad \exists R \in O(n) \text{ such that } \nu = R_{\#}\mu.$$

The quotient space under this relation is denoted by

$$\mathcal{Q}_R := \mathcal{P}_p(\mathbb{R}^d)/\sim_R,$$

where each element $[\mu]_R$ represents the equivalence class (orbit) of $\mu$ under all rotations and reflections.

**Definition C.1** (Rotation-invariant Wasserstein distance). For any two equivalence classes $[\mu]_R, [\nu]_R \in \mathcal{Q}_R$, we define

$$W_p^{(R)}([\mu]_R, [\nu]_R) := \inf_{R \in O(n)} W_p(\mu, R_{\#}\nu).$$

The following result establishes that this construction yields a valid metric on the quotient space.

**Proposition C.2** (Well-defined metric on the rotation quotient space). *The function $W_p^{(R)}$ defines a real metric on $\mathcal{Q}_R$. Moreover, the infimum in the definition is attained.*

*Proof.* First, $W_p^{(R)}$ is well-defined since for any $\mu', \nu'$ in the same equivalence classes as $\mu, \nu$, there exist $R_0, S_0 \in O(n)$ such that $\mu' = R_{0\#}\mu$ and $\nu' = S_{0\#}\nu$. Using the invariance of $W_p$ under orthogonal transformations,

$$\inf_R W_p(\mu', R_{\#}\nu') = \inf_R W_p(R_{0\#}\mu, (RS_0)_{\#}\nu) = \inf_Q W_p(\mu, Q_{\#}\nu),$$

where $Q = R_0^{-1}RS_0$. so the value is independent of the representatives.

The properties of a metric follow directly:

- *Non-negativity and symmetry:* Inherited from $W_p$, since $R \mapsto R^{-1}$ is a bijection on $O(n)$.

- *Identity of indiscernibles:* If $W_p^{(R)}([\mu]_R, [\nu]_R) = 0$, then there exists a sequence $R_k \in O(n)$ with $W_p(\mu, (R_k)_{\#}\nu) \to 0$. Compactness of $O(n)$ ensures a convergent subsequence $R_{k_\ell} \to R^*$, and continuity of the pushforward implies $\mu = R_{\#}^*\nu$. Thus $[\mu]_R = [\nu]_R$. The converse is immediate.

- *Triangle inequality:* For any $\mu, \nu, \eta$ and $R, S \in O(n)$,

$$W_p(\mu, (RS)_{\#}\eta) \leq W_p(\mu, R_{\#}\nu) + W_p(R_{\#}\nu, (RS)_{\#}\eta) = W_p(\mu, R_{\#}\nu) + W_p(\nu, S_{\#}\eta).$$

  Taking the infimum over $R, S$ yields

$$W_p^{(R)}([\mu]_R, [\eta]_R) \leq W_p^{(R)}([\mu]_R, [\nu]_R) + W_p^{(R)}([\nu]_R, [\eta]_R).$$

Finally, since $O(n)$ is compact and $R \mapsto W_p(\mu, R_{\#}\nu)$ is continuous, the infimum is attained. Hence, $W_p^{(R)}$ defines a real metric on $\mathcal{Q}_R$. $\qquad\square$

## C.2 Non-convexity of the optimization over rotations

Although the rotation-induced distance $W_p^{(R)}$ defines a valid metric on the quotient space $\mathcal{Q}_R$, the corresponding optimization problem over rotations is generally non-convex. The following example illustrates this behavior even in a simple two-dimensional case.

**Proposition C.3** (Non-convexity of $W_p(\mu, R_{\#}\nu)$ with respect to rotation). *Consider the cost function $c(x, y) = \|x - y\|_p^p$ with $p \geq 1$. Let the source and target distributions $\mu$ and $\nu$ be defined on $\mathbb{R}^2$ as*

$$\mu = \tfrac{1}{3} \sum_{k=0}^{2} \delta_{u_k}, \qquad \nu = \tfrac{1}{3} \sum_{k=0}^{2} \delta_{u_k},$$

*where $u_k = (\cos(2\pi k/3), \sin(2\pi k/3))$ for $k = 0, 1, 2$. For each rotation matrix $R_\theta$ with angle $\theta \in [0, 2\pi)$, define*

$$f(\theta) := W_p^p\big(\mu, (R_\theta)_{\#}\nu\big).$$

*Then $f(\theta)$ is a non-convex function on $[0, 2\pi)$, and it possesses three disconnected global minimizers.*

*Proof.* Since both $\mu$ and $\nu$ have equal discrete masses, the optimal coupling matches the three support points under cyclic permutations $\sigma_m(k) = k + m \bmod 3$, for $m \in \{0, 1, 2\}$. For any fixed permutation $\sigma_m$, the transport cost is

$$\frac{1}{3} \sum_{k=0}^{2} \|u_k - R_\theta u_{\sigma_m(k)}\|_p^p = \big(2 - 2\cos(\theta - \tfrac{2\pi m}{3})\big)^{p/2},$$

since for unit vectors $a, b$ separated by an angle $\delta$, $\|a - b\|_2^2 = 2 - 2\cos\delta$. Therefore, taking the minimum over the three possible permutations yields

$$f(\theta) = \min_{m \in \{0,1,2\}} \big(2 - 2\cos(\theta - \tfrac{2\pi m}{3})\big)^{p/2}.$$

It follows that $f(\theta)$ achieves its minimum value $f(\theta^*) = 0$ at $\theta^* \in \{0, 2\pi/3, 4\pi/3\}$, corresponding to perfect rotational alignment. Between any two minima (e.g., between 0 and $2\pi/3$), $f$ attains a local maximum at $\theta = \pi/3$, where

$$f(\pi/3) = \big(2 - 2\cos(\pi/3)\big)^{p/2} = 1.$$

Thus, $f$ has a periodic multi-well structure with three disconnected global minima separated by higher-cost regions. Hence, $f(\theta)$ is non-convex, even in this simple discrete case. $\qquad\square$

This example shows that, while the rotation-invariant Wasserstein distance $W_p^{(R)}$ defines a valid metric on the quotient space $\mathcal{Q}_R$, the corresponding optimization problem is generally non-convex. Consequently, finding the optimal rotation $R^*$ that minimizes $W_p(\mu, R_{\#}\nu)$ may require nonconvex optimization strategies or carefully designed initialization schemes.

# D  Additional Experiment Results

## D.1  Additional Experiments for the $RW_2$-Sinkhorn Algorithm

To further assess the $RW_2$-Sinkhorn algorithm, we evaluate its performance on three additional source–target distribution pairs: (1) Gaussian $\to$ Gaussian, (2) Gaussian $\to$ Geometric, and (3) Gaussian $\to$ Poisson. All tests use 1,024 samples, translation magnitude ranges over $\{1, 2, 4, 8, 16\}$, and dimension $d \in \{2, 10\}$, following the setup in Subsection 5.1.2. We report the numerical error (relative to LP ground truth) and the running time.

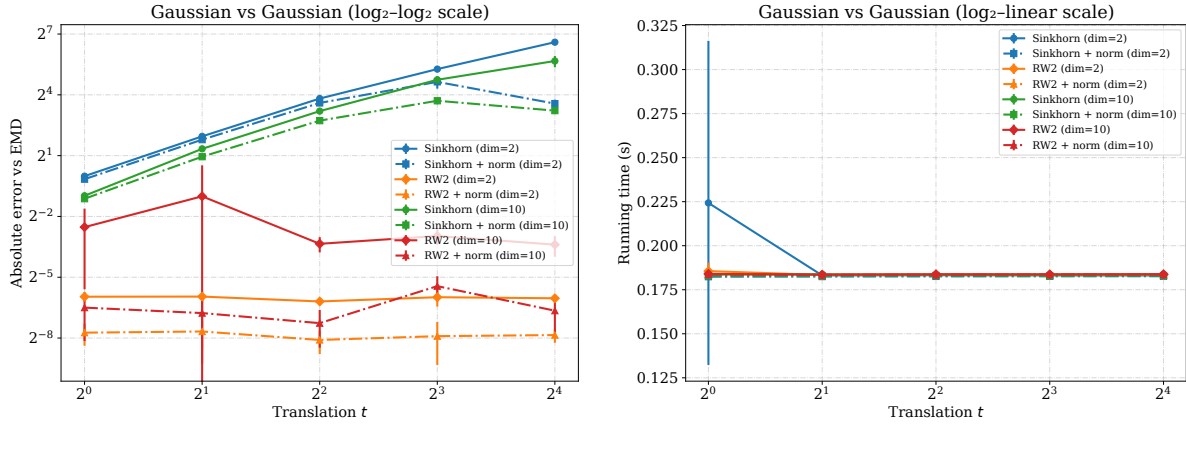

(a) Computation error vs. translation magnitude.

(b) Running time vs. translation magnitude.

Figure 8:  Gaussian $\to$ Gaussian experiment. (a) The $RW_2$-Sinkhorn algorithms achieve lower numerical errors, particularly as the translation magnitude increases. The $RW_2$-Sinkhorn curves are shown at the bottom of the plot, indicating consistently lower numerical errors across different dimensions. Furthermore, the normalization curves lie below their corresponding standard curves, suggesting that both centering and normalization help reduce errors. (b) Running time remains comparable across dimensions.

**Gaussian $\to$ Gaussian.**  In this setting, Figure 8(a) shows that the $RW_2$-Sinkhorn algorithms achieve lower numerical errors, particularly as the translation magnitude increases. The $RW_2$-Sinkhorn curves are shown at the bottom of the plot, indicating consistently lower numerical errors across different dimensions. Furthermore, the normalization curves lie below their corresponding original curves, suggesting that both centering and normalization can help reduce errors. Figure 8(b) shows that the overall running time remains comparable across all dimensions.

**Gaussian $\to$ Geometric.**  For this task, Figure 9(a) shows that the $RW_2$-Sinkhorn algorithms achieve lower numerical errors, as the translation magnitude increases. The $RW_2$-Sinkhorn curves are shown at the bottom of the plot, indicating consistently low numerical errors across different dimensions. Furthermore, the normalization curves lie below their corresponding original curves, suggesting that both centering and normalization can reduce errors. Figure 9(b) shows that applying centering or normalization may introduce a slight increase in running time; however, the overall running time remains comparable.

**Gaussian $\to$ Poisson.**  Figure 10(a) shows that the $RW_2$-Sinkhorn algorithms can achieve lower numerical errors, as the translation magnitude increases. The $RW_2$-Sinkhorn curves are shown at the bottom of the plot, indicating consistently low numerical errors across different dimensions. Furthermore, the normalization curves lie below their corresponding standard curves, suggesting that both centering and normalization can reduce errors. Figure 10(b) shows that applying centering or normalization may introduce a slight increase in running time; however, the overall running time remains comparable.

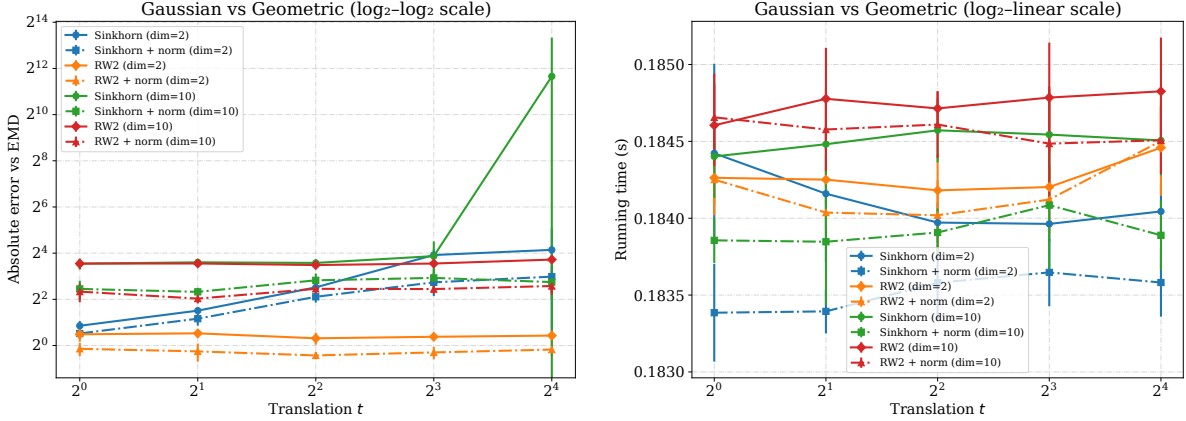

(a) Computation error vs. translation magnitude.

(b) Running time vs. translation magnitude.

Figure 9: Gaussian $\rightarrow$ Geometric experiment. (a) The $RW_2$-Sinkhorn algorithms achieve lower numerical errors, particularly as the translation magnitude increases. The $RW_2$-Sinkhorn curves are shown at the bottom of the plot, indicating consistently low numerical errors across different dimensions. Furthermore, the normalization curves lie below their corresponding original curves, suggesting that both centering and normalization can reduce errors. (b) Applying centering or normalization may introduce a slight increase in running time; however, the overall running time remains comparable.

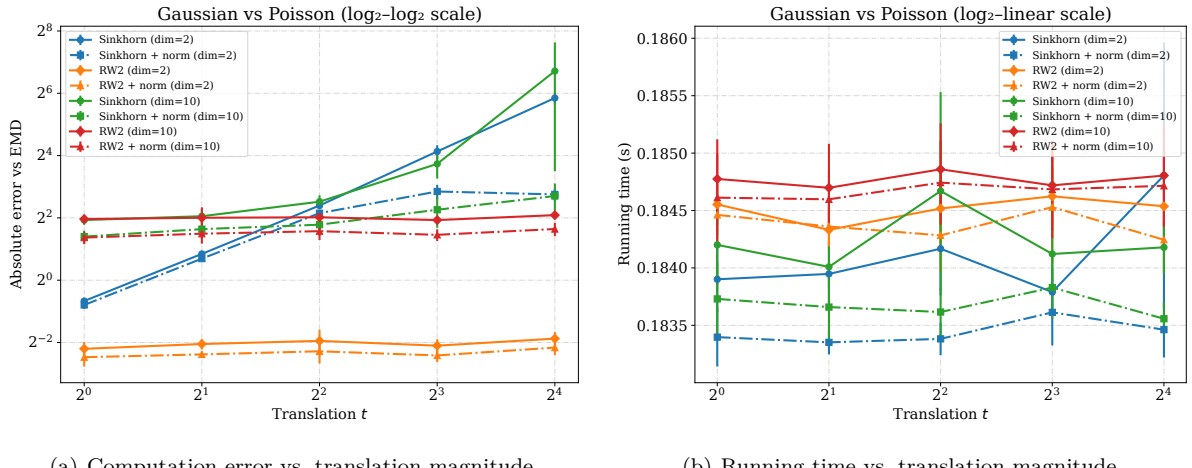

(a) Computation error vs. translation magnitude.

(b) Running time vs. translation magnitude.

Figure 10: Gaussian $\rightarrow$ Poisson experiment. (a) The $RW_2$-Sinkhorn algorithms achieve lower numerical errors, particularly as the translation magnitude increases. The $RW_2$-Sinkhorn curves are shown at the bottom of the plot, indicating consistently low numerical errors across different dimensions. Furthermore, the normalization curves lie below their corresponding original curves, suggesting that both centering and normalization can reduce errors. (b) Applying centering or normalization may introduce a slight increase in running time; however, the overall running time remains comparable.

## D.2 Sequence retrieval experimental results for Section 5.2

**Sequence settings.** In addition to snapshot retrieval, we also extend our evaluation to *sequence retrieval*, where each sequence consists of a temporal series of thunderstorm snapshots. In this experiment, each sequence spans one hour and contains six consecutive snapshots. The similarity between two sequences is measured by the average of the distances computed between corresponding snapshot pairs. Under this setting,

we only report the top-3 sequence retrieval results for all five metrics considered in Section 5.2, namely $\ell_2$, $W_2$, and $RW_p$ with $p \in \{1, 2, 4\}$. Due to the long running time, we do not include other metrics, such as the Gromov–Wasserstein ($GW$) or entropy-regularized Gromov–Wasserstein ($EGW$) distances as baselines.

**Sequence retrieval results.** Figures 11–15 present the sequence retrieval results for the five metrics. For each metric, the results are visualized in four rows: the top row corresponds to the reference thunderstorm sequence, while the second through fourth rows show the top three retrieved sequences identified by the corresponding metric. To increase the diversity of retrieval results, temporally redundant matches (within a 24-hour window) are removed, and only the closest match is retained within each window.

Across the five metrics, clear differences in retrieval behavior can be observed. The classical $W_2$ distance tends to favor sequences that are spatially close to the reference, even when their internal storm structures and temporal evolution differ. In contrast, the proposed $RW_p$ distances consistently emphasize similarity in storm morphology and evolution patterns, leading to retrieved sequences that are visually more similar to the reference sequence. Among the proposed variants, $RW_2$ provides the most balanced performance, yielding sequences that closely match both the shape and temporal progression of the reference thunderstorm. These observations are consistent with the snapshot-based retrieval results reported in Section 5.2 and further demonstrate the similarity of the proposed $RW_p$ distances for spatio-temporal retrieval tasks.

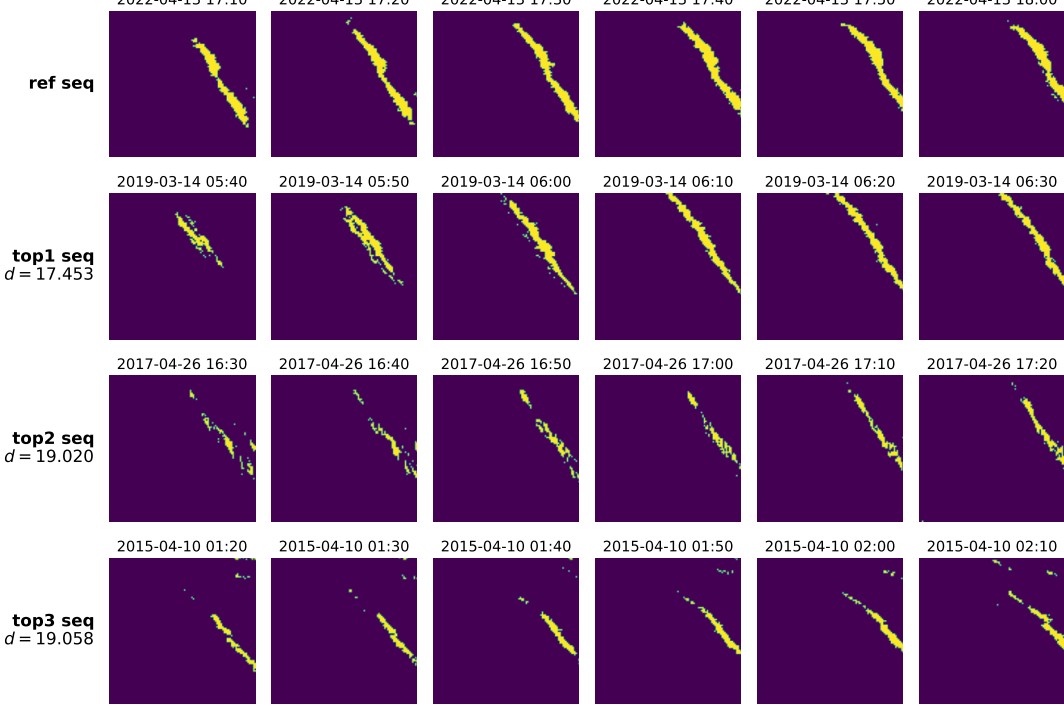

Figure 11: Sequence retrieval results using the $\ell_2$ distance. The first row shows the reference thunderstorm sequence consisting of six consecutive snapshots. The second to fourth rows show the top three retrieved sequences. Each column corresponds to a 10-minute update, and timestamps are shown above each frame.

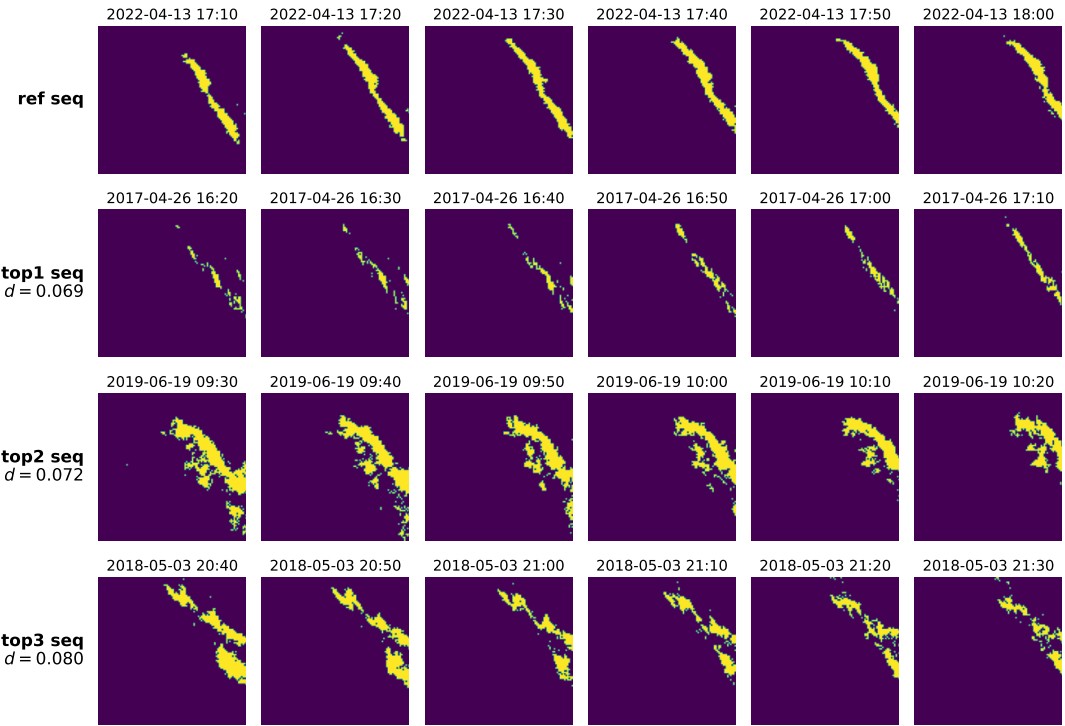

Figure 12: Sequence retrieval results using the $W_2$ distance. While the retrieved sequences are spatially close to the reference, their internal storm structures may differ. The first row is the reference sequence, followed by the top three retrieved sequences from distinct calendar dates.

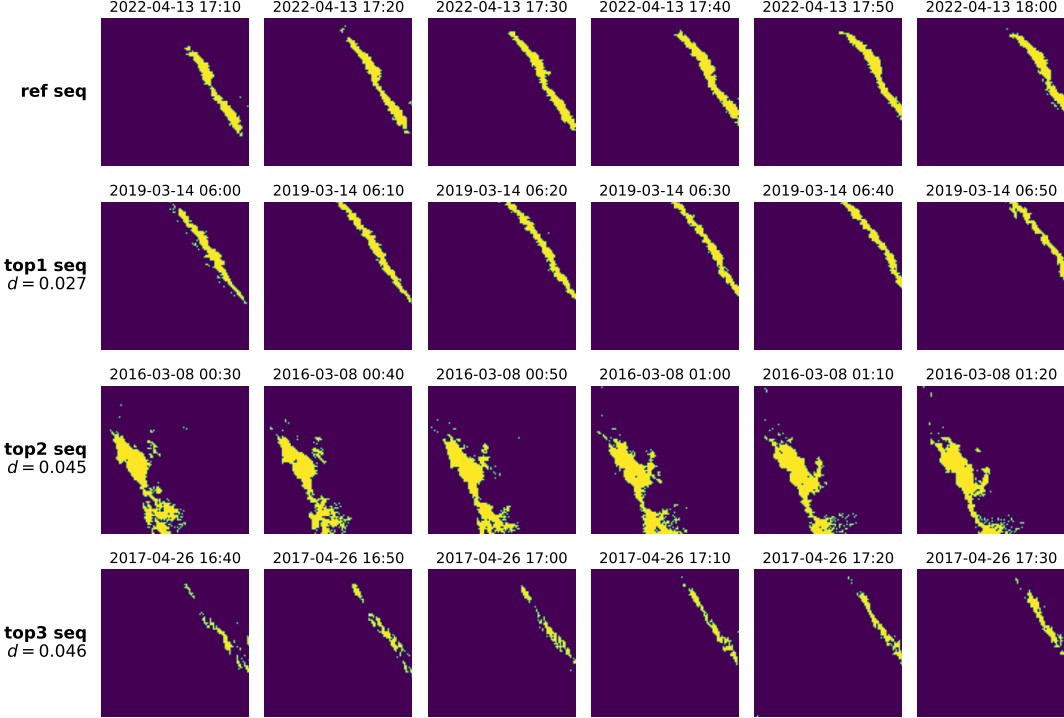

Figure 13: Sequence retrieval results using the $RW_1$ distance. Compared to $W_2$, the retrieved sequences exhibit improved consistency in storm morphology and evolution patterns.

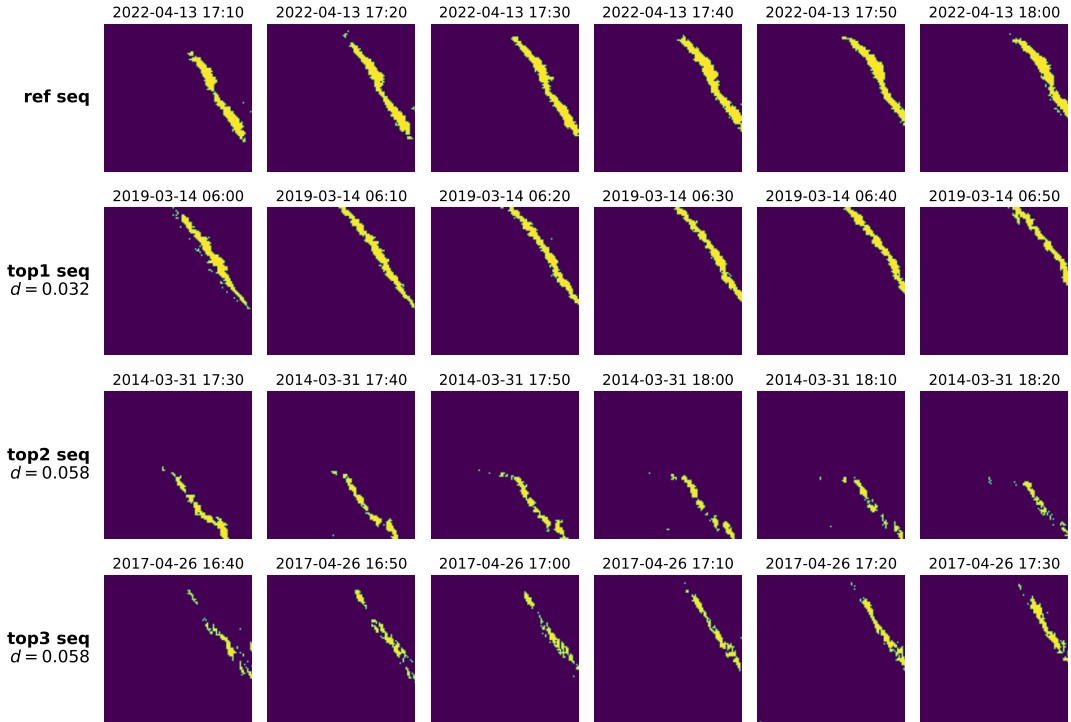

Figure 14: Sequence retrieval results using the $RW_2$ distance. The retrieved sequences closely match the reference sequence in both spatial structure and temporal evolution. This metric provides the most balanced performance among all considered distances.

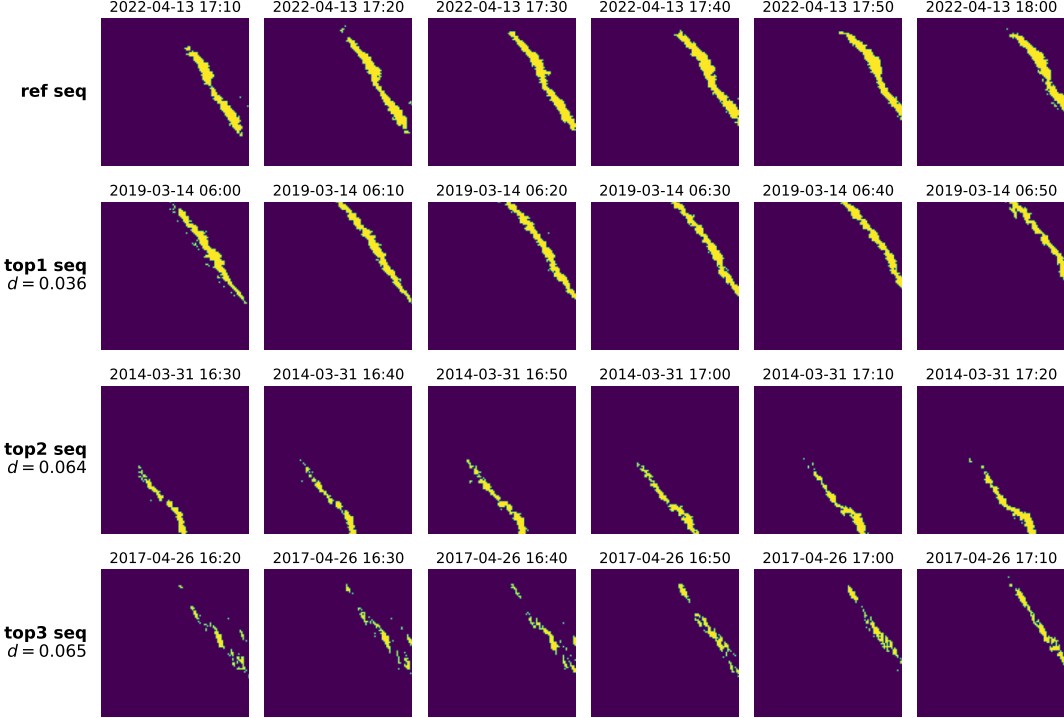

Figure 15: Sequence retrieval results using the $RW_4$ distance. Although slightly more sensitive to outliers, $RW_4$ still preserves the overall storm evolution patterns across time.

