# OpenReview forum: "Relative Translation Invariant Wasserstein Distance"
_TMLR — Accepted by TMLR_

### Review · Reviewer_YRj9 · 2026-03-01

**Summary Of Contributions:**

This paper introduces a relative translation invariant Wasserstein distance. The authors establish that it is a metric on the quotient space of probability distribution up to translation. To compute it in the general case, they propose an alternating optimization algorithm, and show that it converges towards a local minimum.

For an Euclidean cost and p=2, they show that the problem is equivalent to the classical optimal transport problem, and find the optimal translation (which is the difference of the mean of both distributions, which is equivalent with centering both distributions). In particular, the optimal coupling in this case is the same as for the Wasserstein distance. They use this property to provide a more stable algorithm to compute the OT problem, either with the Linear Program or the entropic approximation and Sinkhorn algorithm.

Finally, they demonstrate that their algorithm has lower computational error than directly solving the OT problem for Euclidean cost and p=2 on several toy datasets. And they provide an application on a thunderstorm pattern retrieval task, showing that the relative translation invariant distance has better results than the OT problem, and that for p=1, the results are more robust to noise.

**Additional Comments:**

Overall, the introduction and analysis of the relative translation invariant Wasserstein distance are sound. In particular, the algorithm is shown to converge to stationary points, and is thus meaningful to compute the distance. Moreover, the pattern retrieval task seems to be large scale, which demonstrates that the method can be used practice. Nonetheless, I am not convinced by the results and applications for $p=2$. And I believe that more motivations for $p\neq 2$ would improve the work.

**Audience:**

Yes

**Audience Explanation:**

Optimal transport is of interest to TMLR's audience, and some of the findings of this paper may be of interest.

**Claims And Evidence:**

No

**Claims Explanation:**

The claims made in the submission are generally supported by sound evidence. Nonetheless, in its current form, the paper lacks clarity. Moreover, its connection to the existing literature could be strengthened, and the experimental results could be compared with more relevant methods.

**Requested Changes:**

I think several changes are needed. First, it would be good to improve the readibility of the paper. Second, some of the results are related to known results, which is not underlined. Also, the review of related works could be improved and the experimental section could also be improved by adding comparisons with more related methods.

**Results on $RW$.**

The sentence "Corollary 3.4 generalizes the classical Wasserstein–Bures metric" is not clear to me. I guess what is meant here is that $RW_2$ generalizes the Bures distance (as the Bures distance is the distance between the covariance matrices, hence between centered Gaussian). It would be better to recall what are the Bures and Bures-Wasserstein distances in Section 2.

Proposition 3.2 and Corollary 3.4 come from the classical property that the squared 2-Wasserstein distance is equal to the squared Euclidean distance between the means + the squared 2-Wasserstein distance between centered distributions (see e.g. Remark 19 in [1]). These results are the same, as the optimal translation is $\bar{\mu}-\bar{\nu}$, which comes back to center both distribution (or shift $\mu$ to have the same expectation as $\nu$). It should be underlined more clearly.

In Section 3.2, it is written that "ROT problem can be solved analytically when dimension d = 1", but the analytical results are not provided. For $d\ge 2$, it can also be solved in the Gaussian case, but it is not clearly written.


**Related works.**


Papers which are very related are not discussed (there is a Related work Section, which is very generic, and does not mention works which are actually related). For example, there is no mention of [2], which focuses on an invariant OT problem. In particular, their equation (7) seems to be more general than the relative-translation invariant OT problem introduced in this work. Also, another OT problem which is also translation invariant is the Gromov-Wasserstein problem (with Euclidean distance), but it is not discussed nor compared with in the pattern retrieval task (while it could be an easy to beat baseline in term e.g. of running time). The paper introduces in appendix a rotation invariance metric, taking the inf over rotations. This is also related to what is done in equation (6) of [3].


Page 2, I don't think that it makes sense to cite (Liero et al, 2018) and (Janati et al, 2020) for Information Geometry as these papers talk about unbalanced optimal transport.

The reference "Rajendra N. K., Nicolas Courty, and Rémi Flamary. Wasserstein–bures metric for gaussian measures. SIAM Journal on Imaging Sciences, 12(4):2311–2341, 2019." does not seem to exist. Also, I did not find "Yongxin Chen, Tryphon T. Georgiou, and Michele Pavon. Optimal transport in the space of gaussian measures. IEEE Transactions on Automatic Control, 63(9):2913–2928, 2018.".


**Experiments.**

The main motivation of using the property of Corollary 3.4 to compute $W_2^2$ or its entropic regularization comes from the fact that it will allow to avoid large values of the cost matrix. While it is reasonable, a common method which is used in general is to divide the cost matrix by the max. So, I am not sure that centering the distributions would be more efficient than simply dividing the cost by the maximum. For instance, we could take two distributions with the same expectation, but one with bigger variance, and we would still have large values in the cost matrix. Hence, it would be good to discuss and add comparisons with this classical normalization.


The pattern retrieval task could be compared with the Gromov-Wasserstein distance, which is also invariant to translation.


Typos:
- Page 2: "achieve significantly numerical stability"
- Page 2, 7: "Wasserstein–Bures metric" (not consistent with abstract where it is called the Bures-Wasserstein metric)
- Page 2: "we extend Gaussian distributions to a much broader class of distributions"
- Page 2: (ichi Amari, 2016) -> (Amari, 2016)
- Page 3: "Equation equation 1"
- Page 4: "Equation equation 2"
- Page 6: "Equation equation 3"


[1] Peyré, G., & Cuturi, M. (2019). Computational optimal transport: With applications to data science. Foundations and Trends® in Machine Learning, 11(5-6), 355-607.

[2] Alvarez-Melis, D., Jegelka, S., & Jaakkola, T. S. (2019, April). Towards optimal transport with global invariances. In The 22nd International Conference on Artificial Intelligence and Statistics (pp. 1870-1879). PMLR.

[3] Vayer, T., Flamary, R., Courty, N., Tavenard, R., & Chapel, L. (2019). Sliced gromov-wasserstein. Advances in Neural Information Processing Systems, 32.

---

> ### Author Response · Authors · 2026-03-15
> **Improved Clarity and Expanded Comparisons with Related Methods**
>
> We appreciate the reviewer’s assessment. We interpret this comment as indicating that the theoretical results are sound but the presentation and comparisons can be improved. In the revision, we improved the clarity of the theoretical sections (especially the decomposition results and their relation to classical Wasserstein results) and strengthened the experimental section by adding additional baselines and comparisons, including normalization techniques for the cost matrix and invariant OT method (the Entropic GW distance). Specifically, the revised manuscript includes the following major updates:
>
> 1. **Expanded related work.**
>    We enriched the related work section by incorporating additional comparison methods, particularly translation-invariant distances and related optimal transport formulations. These include the Gromov–Wasserstein (GW) and entropic Gromov–Wasserstein (EGW) distances, robust optimal transport, the Bures–Wasserstein metric, the Procrustes–Wasserstein distance, and sliced optimal transport methods.
>
> 2. **New Section 2.2.**
>    We added a dedicated subsection introducing the Bures–Wasserstein metric and its relevance to our formulation.
>
> 3. **Revisions to Section 3.2.**
>    We expanded the discussion of the one-dimensional case ($d=1$), with additional detailed results provided in Appendix A.4. We also clarified the treatment of the higher-dimensional ($d \ge 2$) and Gaussian settings.
>
> 4. **Revisions to Section 3.3.**
>    We strengthened the discussion of normalization and clarified its connection to the Bures–Wasserstein metric.
>
> 5. **Updates to Section 5.1 and Appendix D.**
>    We included normalization-based variants as additional baselines in the experimental evaluation.
>
> 6. **Updates to Appendix C.**
>    We added discussion on the relationship between our formulation and the Procrustes–Wasserstein distance as well as sliced optimal transport methods.
>
> 7. **Revisions to Section 5.2.**
>    We incorporated the Gromov–Wasserstein (GW) and entropic Gromov–Wasserstein (EGW) distances as additional baselines.

---

> > ### Author Response · Authors · 2026-03-15
> > **Correction of Bibliographic References**
> >
> > We thank the reviewer for carefully checking these references. The first reference is indeed incorrect and has been removed from the revised manuscript.
> >
> > For the second citation, the reference was mistakenly written. The correct reference we intended to cite is:
> >
> > *Yongxin Chen, Tryphon T. Georgiou, and Michele Pavon. Optimal transport in systems and control. Annual Review of Control, Robotics, and Autonomous Systems, 4:89--113, 2021.*
> >
> > We corrected this reference in the revised version and carefully verified the bibliography to avoid similar issues.

---

> ### Author Response · Authors · 2026-03-15
> **Clarification of the Relation to the Bures–Wasserstein Distance**
>
> We agree that this statement was not sufficiently clear and have revised it accordingly. In the revised manuscript, we added a brief introduction in Section 2.2 and in the related work section to recall the Bures–Wasserstein distance between Gaussian measures.
>
> Specifically, for Gaussian measures $\mu = \mathcal{N}(m_1, \Sigma_1)$ and $\nu = \mathcal{N}(m_2, \Sigma_2)$, the squared $2$-Wasserstein distance admits the closed-form expression,
> $W_2^2(\mu, \nu)=|m_1 - m_2|^2+Tr(\Sigma_1)+Tr(\Sigma_2)-2Tr\left((\Sigma_1^{1/2} \Sigma_2 \Sigma_1^{1/2})^{1/2}\right).$
>
> The covariance term
>
> $d_{BW}^2(\Sigma_1, \Sigma_2)=Tr(\Sigma_1)+Tr(\Sigma_2)-2Tr\left((\Sigma_1^{1/2} \Sigma_2 \Sigma_1^{1/2})^{1/2}\right)$
>
> is known as the *Bures–Wasserstein distance* between covariance matrices.
>
> We then clarified that the $RW_2$ metric can be interpreted as extending this decomposition beyond the Gaussian setting: while the Bures–Wasserstein distance captures the covariance component for Gaussian measures, the $RW_2$ metric generalizes the centered component of the $W_2$ distance to probability distributions with finite moments. We revised the wording around Corollary 3.4 to make this connection explicit and avoid overstating the contribution.

---

> ### Author Response · Authors · 2026-03-15
> **Clarification of the Classical Mean–Centering Decomposition**
>
> We thank the reviewer for pointing out this important connection. In the revised version, we explicitly acknowledged this classical decomposition of the squared $2$-Wasserstein distance and cite the appropriate references in Section 3.3.
>
> Our contribution in the Section 3.3 is to interpret this relationship through the $RW_2$ quotient metric induced by translation equivalence classes and to leverage the resulting invariance of the optimal coupling matrix (in the discrete case) to design more stable algorithms.
> We revised the discussion around Proposition 3.2 and Corollary 3.4 to make this connection clearer and properly acknowledge the classical result.

---

> ### Author Response · Authors · 2026-03-15
> **Clarification of Analytical Results for $d=1$ and the Gaussian Case**
>
> We thank the reviewer for pointing this out. We agree that this statement should be clarified. In the revised version, we added a brief intuitive explanation of cyclical monotonicity and its role in enabling analytical tractability in the one-dimensional setting in Section 3.2. And we also provided a derivation and explicit formulation of the $d=1$ case in Appendix A.4, with clearer cross-referencing from the main text.
>
> We also clarify the Gaussian case for $d \ge 2$, where the problem admits a closed-form solution, and make this discussion explicitly in the revised manuscript.

---

> ### Author Response · Authors · 2026-03-15
> **Expanded Related Work and Connections to Invariant OT Methods**
>
> We agree and have substantially expanded the related work section. In particular:
>
> * We discussed invariant optimal transport formulations that optimize over transformations and related them to our formulation of translation-invariant OT.
> * We discussed the relation with Gromov–Wasserstein distances, which can also be invariant to translations when Euclidean distances are used.
> * We briefly discussed robust optimal transport formulations.
> * We clarified the relation with the Bures–Wasserstein metric, which arises in the Gaussian setting as the covariance component of the $2$-Wasserstein distance and defines a Riemannian geometry on covariance matrices.
> * We connected our rotation-invariant extension (Appendix C) to prior work on invariance and sliced Gromov–Wasserstein methods.
>
> ---

---

> ### Author Response · Authors · 2026-03-15
> **Revised Citations on Information Geometry**
>
> We thank the reviewer for pointing this out. We agree that these works primarily concern unbalanced optimal transport and that citing them in the context of information geometry may be misleading. In the revised manuscript, we remove them and adjust the discussion accordingly.

---

> ### Author Response · Authors · 2026-03-15
> **Comparison with Cost Matrix Normalization**
>
> We thank the reviewer for pointing out this important baseline and for the helpful suggestion. We agree that normalization of the cost matrix (e.g., dividing by its maximum value and rescaling) is a commonly used stabilization strategy in practice.
>
> To clarify the distinction, we added a discussion in Section 3.3 emphasizing that centering and normalization address different numerical issues. In particular, centering primarily mitigates instability caused by large mean offsets between distributions, whereas normalization reduces scale differences in the cost matrix. As noted by the reviewer, large variance can still lead to large cost values even after centering, and therefore normalization remains a complementary technique.
>
> To provide a more comprehensive empirical comparison, we revised the experiments in Section 5.1 and Appendix D to include the following settings:
>
> * Standard OT solvers,
> * OT solvers with cost normalization (by the maximum entry),
> * The proposed $RW_2$-based stabilization method,
> * OT solvers combining both normalization and the proposed $RW_2$ stabilization.
>
> These additions clarify the respective roles of centering and normalization and demonstrate that the proposed approach provides complementary benefits in practice.

---

> ### Author Response · Authors · 2026-03-15
> **Comparison with Gromov–Wasserstein distances**
>
> We thank the reviewer for the suggestion. In the revised manuscript, we added  Gromov--Wasserstein (GW) and Entropic Gromov--Wasserstein (EGW) as additional baselines in the thunderstorm pattern retrieval experiment and report both retrieval accuracy and computational cost.
> The detailed results can be found in section 5.2.

---

> ### Author Response · Authors · 2026-03-15
> **Reference and Typographical Corrections**
>
> We thank the reviewer for pointing this out. We carefully verify and correct all references and address the listed typos in the revision.
> We also ensure consistent terminology (e.g., using Bures--Wasserstein rather than Wasserstein--Bures) and correct citation formatting throughout the manuscript.
> The typo ``Equation equation'' is caused by an incorrect use of \texttt{\textbackslash eqref}, and we have carefully reviewed the manuscript and corrected those issues.

---

> ### Comment · Reviewer_YRj9 · 2026-03-18
>
> Thank you for your answer and for addressing my comments. I find the paper much clearer now, and more complete.
>
> Two additional comments:
>
> - For the Gaussian case, you are calling the distance between covariances the Bures-Wasserstein distance, while the correct terminology should be the Bures distance (see e.g. [1]), and the full distance between Gaussian can be called either Wasserstein or Bures-Wasserstein distance.
> - Maybe a natural additional baseline for the Thunderstorm Pattern Retrieval task would be to compute the $W_p$ distance between centered distributions (for $p=1$ and $4$). It would show the benefit of actually computing $RW_p$ with the alternate optimization scheme.
> - Maybe I missed it, but I did not see which cost you are using for GW and EGW (I guess the squared Euclidean distance).
>
> Typos:
> - page 11: "is draw from"
>
> [1] Bhatia, R., Jain, T., & Lim, Y. (2019). On the Bures–Wasserstein distance between positive definite matrices. Expositiones mathematicae, 37(2), 165-191.

---

> > ### Author Response · Authors · 2026-03-19
> > **Further Revisions**
> >
> > We sincerely thank the reviewer for the positive feedback and for the additional helpful comments.
> >
> > ### **Terminology for Gaussian distances**
> >
> > We thank the reviewer for this helpful remark. We agree that the distance between covariance matrices is commonly referred to as the **Bures distance**, while the full distance between Gaussian distributions is typically called the **Wasserstein distance**, and in some parts of the literature also the **Bures–Wasserstein distance**.
> >
> > At the same time, we note that the terminology is not entirely uniform across the literature. In particular, some references [1,2] use the term *Bures–Wasserstein distance* to denote metrics on covariance matrices or distances associated with zero-mean Gaussian distributions, rather than the full Gaussian transport distance. To avoid ambiguity, we have revised the manuscript to clearly distinguish these notions.
> >
> > We would be grateful if the reviewer could kindly confirm that this clarification aligns with their intended interpretation.
> >
> > **References**
> >
> > [1] van Oostrum, J. *Bures–Wasserstein geometry for positive-definite Hermitian matrices and their trace-one subset.* Information Geometry, 5, 405–425 (2022).
> >
> > [2] Haasler, I., & Frossard, P. *Bures–Wasserstein Means of Graphs.* AISTATS (2023).
> >
> > ---
> >
> > ### **Centered-distribution baseline**
> >
> > We thank the reviewer for this valuable suggestion. In the revised manuscript, we include additional baselines that compute Wasserstein distances between centered distributions. This comparison helps isolate the role of translation invariance and further highlights the benefit of solving the proposed $RW_p$ problem via the alternating optimization scheme.
> >
> > ---
> >
> > ### **Cost function for GW and EGW**
> >
> > We apologize for the lack of clarity. In both $GW$ and $EGW$ experiments, we use the **Euclidean distance** as the ground cost. This has now been explicitly stated in the experimental section 5.2.1.
> >
> > ---
> >
> > ### **Typo**
> >
> > We thank the reviewer for pointing out the typo on page 11. It has been corrected to *“is drawn from.”*
> >
> > ---
> >
> > We appreciate the reviewer’s careful reading and constructive suggestions, which have helped improve the clarity and completeness of the paper.

---

> > > ### Comment · Reviewer_YRj9 · 2026-03-19
> > >
> > > Thank you for addressing my comments.
> > >
> > > - I guess it is ok to call the Bures distance the Bures-Wasserstein distance when only centered distributions are considered. But when also non centered Gaussian are considered, it might be more clear to make the distinction between them.
> > > - Thank you for adding the centered baseline. The results are very close to $RW_p$, even though there is a small improvement for $RW_p$. It seems to me that the computational overhead of $RW_p$ may not be worth it for large scales computations in practice. Nonetheless, it is an interesting result.

---

> ### Author Response · Authors · 2026-03-20
> **Further Revisions**
>
> We sincerely thank the reviewer for the careful follow-up and for the constructive suggestions.
>
> **Terminology for Gaussian transport distances.**
> We agree that when both centered and non-centered Gaussian distributions are considered, it is important to clearly distinguish between the Bures metric on covariance matrices and the full Wasserstein distance between Gaussian measures. To improve clarity, we have added a footnote in Section 2.2 of the revised manuscript explicitly explaining this distinction and clarifying our terminology. In particular, we use the term *Bures metric* when referring to the covariance component, while the full transport distance between Gaussian distributions is referred to as the Wasserstein distance.
>
> **Centered baseline and computational trade-off.**
> We also thank the reviewer for the positive remark on the inclusion of the centered baseline. We agree that the performance is very close. In our formulation, the initialization $t = \bar{\nu} - \bar{\mu}$ in $RW_p$ implicitly corresponds to a centering step between the two distributions. As a result, a trade-off arises: $RW_p$ is expected to require additional computational time compared to explicitly centered baselines such as $W_{1c}$ or $W_{4c}$. At the same time, the iterative refinement in $RW_p$ may lead to slightly improved alignment and thus more refined retrieval results.
>
> Due to the sparsity of the current dataset, we are currently limited to the experiments reported in the manuscript. A more extensive empirical study at larger scales would help further clarity this trade-off between computational overhead and performance gains.
>
> We thank the reviewer again for these insightful comments and for recognizing the potential interest of the proposed approach.

---

### Review · Reviewer_3TX4 · 2026-03-02

**Summary Of Contributions:**

This paper focuses on a new variant of Wasserstein distances, called "Relative translation invariant Wasserstein" distances (noted $RW_{p}$). The authors' motivation is inducing a translation-invariant metric (in the measure theoretic sense) to be more robust to noises appearing in data. I generally agree with the authors assessment of their contributions, meaning:

1. A new family of metrics, meaning, the $RW_{p}$, $p \in [1, +\infty)$
(2, 3). A new algorithm for computing these metrics (and the translation-invariant transport plan, for that matter),
4. The demonstration that the $RW_{p}$ works in an empirical setting.

__The main strength__ of this paper is that it is a nice engineering of the original OT problem.

__The main weakness__ of this paper is the way they present their empirical findings. See requested changes below.

**Audience:**

Yes

**Audience Explanation:**

Probability metrics are of general interest to the TMLR community. Furthermore, I think that the proposed metric can be applied to a broad set of Machine Learning problems. The application in this paper (remote sensing) is also of interest to the community.

**Claims And Evidence:**

No

**Claims Explanation:**

The last question is a binary yes/no. I would actually answer _partially_. While I think the authors do a good job demonstrating the computational aspects of their work (Section 5.1 in special), I think section 5.2 could be better articulated.

There are a couple of issues with Section 5.2,

1. The main comparison and analysis in this section comes from Figure 4, which seems to be qualitative. The authors include a $d = ...$ on top of each retrieval, but it is not clear what this numeric value refers to. I suppose it refers to the metric used in each row (e.g. $\ell_{2}$, $W_{2}$, etc), but that would mean the values are not really comparable. In that sense, other than a qualitative comparison, we can't really say that the $RW_{p}$ is doing a better job at retrieval.

2. From the introduction, and the authors' paper overall, I was expecting some analysis into why translation invariance is important for the thunderstorm pattern retrieval. For instance, do we expect the noise of sensors in that particular problem to be additive (hence inducing a translation in the images)?

**Requested Changes:**

1) On top of the qualitative results of figures 4 and 10 through 14, the authors should include quantitative results on their retrieval task. Here, I think accuracy and precision-recall curves could be applied.

2) All plots relating quantities should be in log-log scale, especially Figures 1 (a), 1 (b), 2 (a), 2 (b), 3 (a) and 3 (b). I think the same could be applied to the similar figures in the appendix.

---

> ### Author Response · Authors · 2026-03-15
> **Log–Log Plot Revisions**
>
> We thank the reviewer for this helpful suggestion. We agree that log–log scaling can improve the readability and interpretation of these plots. In the revised manuscript, we updated most of Figures 1–3 (as well as the corresponding figures in the appendix) to use log–log scales where appropriate.
>
> For Figures 1(b), 3(b), and the corresponding running-time plots in the appendix, we use a logarithmic scale only on the $x$-axis while keeping the $y$-axis linear, since we found that the running time are relatively similar and a log scale on the $y$-axis diminished the readability of tick marks and overall visual clarity.

---

> ### Author Response · Authors · 2026-03-18
> **Quantitative Evaluation of Retrieval Performance**
>
> We thank the reviewer for this helpful suggestion and agree that quantitative evaluation can further strengthen the retrieval experiments.
> In the revised manuscript, we included a quantitative retrieval evaluation based on temporal consistency in the thunderstorm sequence in Section 5.2.2.
>
> **Motivation.**
>
> Our retrieval task was primarily motivated by practical meteorological analysis, where similarity between storm systems is often assessed in terms of their morphology and structural evolution.
> This makes it a natural application domain for translation-invariant similarity measures such as the proposed $RW_p$ distances.
> In addition, radar observations are subject to multiple sources of uncertainty, including sensor noise, temporal acquisition latency, and other measurement perturbations.
> The use of $RW_p$ distances can help mitigate the impact of such uncertainties.
> A relative translation-invariant transport metric is therefore beneficial, as it can improve the robustness of similarity assessment under these perturbations.
>
> **Lack of annotated datasets.**
> Even though supervised quantitative is necessary, we acknowledge that fully supervised quantitative evaluation is challenging in this case.
> To the best of our knowledge, there is currently no publicly available large-scale thunderstorm dataset with human-annotated labels that explicitly reflect structural similarity between storm patterns (e.g., derived from MRMS dataset or other related radar datasets).
> Furthermore, human annotations may not always objectively capture distributional similarity.
> This limitation explains why quantitative retrieval accuracy was not included in the original submission.
>
> **Weakly supervised evaluations.**
> To enable quantitative assessment, we adopt a weakly annotation protocol that leverages the natural temporal structure of the data.
> Specifically, given a temporal threshold $T_0$, for a reference snapshot at time $t$, we annotate snapshots within the temporal window $[t-T_0,\, t+T_0]$ as the *similar* category, and snapshots outside this window as *dissimilar*.
> This labeling assumption is reasonable, since thunderstorm systems typically evolve continuously over short time scales, and temporally adjacent snapshots are therefore more likely to share similar spatial structures.
> We acknowledge that this constitutes a weak supervision strategy, as temporal proximity does not always guarantee structural similarity and may overlook certain meteorological factors.
>
> **Translation enhancement and $\lambda_m$ tuning.**
> The proposed $RW_p$ distances primarily capture morphological similarity and are inherently translation-invariant.
> While this property improves robustness to spatial alignment, it does not explicitly account for the physical displacement of storm systems between sequential radar snapshots.
> When the temporal gap between observations is relatively large (e.g., a radar refresh interval of approximately 10 minutes), such displacement becomes a non-negligible factor in this supervised evaluation.
> To address this limitation, we also incorporate an additional translation regularization term weighted by a coefficient $\lambda_m$, which balances structural similarity and spatial displacement.
> The optimal value of $\lambda_m$ is selected via a linear search over the interval $[0.2,1.6]$, and we empirically find that $\lambda_m = 0.6$ yields the best retrieval performance (see Figure 5(b)).
>
> **Evaluations.**
> Based on this protocol, we report quantitative retrieval results using precision–recall curves (and related metrics) for all compared methods in the revised manuscript.
> More detailed results can be found in Section 5.2.2.

---

### Review · Reviewer_GqXT · 2026-03-09

**Summary Of Contributions:**

The authors propose an extension of Wasserstein distance, encoding invariance to translations. The invariance baked into the distance is well justified and the measure is developed in a rigorous manner, with full theoretical analysis for all aspects. The authors also provide practical algorithms for computing the distance and I was happy to see both Sinkhorn- and LP-based algorithms already in the first paper introducing the new measure. Both high-level explanation and sufficient details of both algorithms are provided, to the degree that a qualified expert should be able to implement these. The empirical results shows significant and impressive improvements, and the experimentation goes well beyond many theoretical papers, including also a real demonstration in the thunderstorm use-cases.

Overall, I think this is a strong paper with no obvious shortcomings. I have not verified the theoretical arguments in detail, but did check the claims and the gist of the analysis.

**Audience:**

Yes

**Audience Explanation:**

The target audience of the paper is somewhat narrow due to rather high degree of theoretical difficulty, but for anyone working on theoretical properties of distribution measures that work is without doubt valuable. The work makes clear contributions for the theoretical line of work, and has potential for influencing broader family of machine learning algorithms that rely on Wasserstein distances in a reasonable time horizon even though the measure is here evaluated only in isolation. Hence the work will be of some interest even for the broader ML audience, and the concrete algorithms will be immediately usable for practitioners and they are explained well. The general interest is further supported by inclusion on sufficiently interesting real world example, the thunderstorm use-case.

**Broader Impact Concerns:**

None.

**Claims And Evidence:**

Yes

**Claims Explanation:**

The authors make clear claims and provide strong support for them, in form of both theoretical proofs and empirical validation. The paper is fairly exemplary in this respect, characterising exactly the claims that are made and directly validating them. The empirical results are very clear, showing effectively a qualitative difference between the proposed measure and the standard Wasserstein distance, not just a small improvement (e.g. Fig. 3).

**Requested Changes:**

My only minor comment concerns Section 3.2 that says $d=1$ can be solved analytically because of cyclical monotonicity. If I understood correctly, you do not consider d=1 anywhere else in the paper. This would call for slightly expanded treatment, like description of what cyclical monotony refers to, as well as explicitly saying that you focus for $d>1$ for the rest of the paper. Maybe add brief description of $d=1$ for Appendix for completeness?

---

> ### Author Response · Authors · 2026-03-15
> **Analytical Characterization of the One-Dimensional ROT Problem**
>
> We thank the reviewer for this helpful suggestion. The reviewer is correct that the one-dimensional case was only briefly discussed in the previous version, while the primary focus of the paper is on higher-dimensional settings.
>
> In one dimension, the ROT problem admits an analytical characterization due to the monotone structure of optimal transport plans implied by cyclical monotonicity. In particular, using the quantile formulation of the $p$-Wasserstein distance, the optimal transport plan corresponds to the monotone rearrangement between the distributions. This reduces the ROT problem to a one-dimensional convex optimization with respect to the translation variable, allowing the optimal translation to be characterized explicitly.
>
> We also note that the phrase “solved analytically” used in the previous version may have been misleading. More precisely, the one-dimensional setting admits an analytical characterization rather than a fully closed-form solution. We have revised this terminology accordingly in the manuscript.
>
> To further improve clarity and accessibility, we made the following revisions:
>
> * Added a brief intuitive explanation of cyclical monotonicity and its role in enabling analytical tractability in the one-dimensional setting (Section 3.2).
> * Explicitly clarified that the main theoretical and algorithmic developments focus on the case $d \ge 2$, where the optimization problem becomes substantially more challenging.
> * Provided a derivation and explicit formulation of the $d = 1$ case in Appendix A.4, with clearer cross-referencing from the main text.
>
> We agree that these revisions improve the clarity and completeness of the presentation.

---

> > ### Comment · Reviewer_GqXT · 2026-04-02
> > **Response**
> >
> > Thank you for the clarifications. I checked the new elements and they address my concerns well, and I have no open questions regarding the paper.

---

### Decision · Action_Editor_P91M · 2026-04-15

**Recommendation:** Accept as is

**Audience:**

Yes

**Audience Explanation:**

The paper propose a novel optimal transport formulation that can be of interest to the TMLR audience and the ML community at large.

**Claims And Evidence:**

Yes

**Claims Explanation:**

The authors propose a novel translation invariant Wasserstein distance and make connection with the Gaussian case. They propose algorithms and evaluate them in numerical experiments. The reviewers originally had some concerns about the writing of the paper, references to other OT formulation and experiments.  But the authors did a detailed reply and included all their comments in the revision that makes the paper stronger and the experiments are now convincing.